# ReLo is a simple and rapid colocalization assay to identify and characterize direct protein–protein interactions

Harpreet Kaur Salgania [1], Jutta Metz [1] & Mandy Jeske [1]✉

The characterization of protein-protein interactions (PPIs) is fundamental to the understanding of biochemical processes. Many methods have been established to identify and study direct PPIs; however, screening and investigating PPIs involving large or poorly soluble proteins remains challenging. Here, we introduce ReLo, a simple, rapid, and versatile cell culture-based method for detecting and investigating interactions in a cellular context. Our experiments demonstrate that ReLo specifically detects direct binary PPIs. Furthermore, we show that ReLo bridging experiments can also be used to determine the binding topology of subunits within multiprotein complexes. In addition, ReLo facilitates the identification of protein domains that mediate complex formation, allows screening for interfering point mutations, and it is sensitive to drugs that mediate or disrupt an interaction. In summary, ReLo is a simple and rapid alternative for the study of PPIs, especially when studying structurally complex proteins or when established methods fail.

The identification and characterization of protein–protein interactions (PPIs) are a routine laboratory practice and lay the foundation for understanding biological processes. PPIs can be identified by several well-established, mass spectrometry-coupled screening methods, including coimmunoprecipitation (co-IP), tandem affinity purification, and proximity-dependent labeling approaches, such as BioID or APEX[1–6]. This results in a list of candidate interacting proteins, which are typically ranked according to their abundance in the eluate fractions. Determining which of these candidates are truly direct binding partners requires subsequent validation experiments, often using in vitro methods, such as GST pull-down assays, which depend upon the availability of purified proteins. When proteins are poorly soluble and cannot be obtained through recombinant protein expression, or when expertise in recombinant protein expression and purification methods is lacking, PPIs can be validated using cell-based assays.

Yeast two-hybrid (Y2H) and protein complementation assays (PCA) are well-established techniques in which an interaction results in the reconstitution and subsequent detection of a split reporter protein, such as a transcription factor, ubiquitin, an enzyme, or a fluorescent protein[7–12]. However, standard Y2H and PCA assays may not be well suited for the analysis of potentially unstable proteins. If these proteins are poorly expressed or rapidly degraded in a cell, this will lead to unreliable, false-negative results. Therefore, to obtain conclusive data, negative results require additional assessment of the protein expression levels, which complicates the process, especially when investigating many PPIs.

Cell-based PPI methods, which are better suited for testing interactions involving potentially unstable proteins are based on fluorescent protein tagging and colocalization readouts, allowing the simultaneous monitoring of both PPIs and protein expression levels by fluorescence microscopy. The readout of these colocalization assays is usually the translocation of a protein upon its association with a second distinctly localized protein (e.g., localization to a membrane, the nucleus, or granules). 'Cytoskeleton-based assay for protein–protein interaction' (CAPPI), 'membrane recruitment assay' (MeRA), and 'knocksideways in plants' (KSP) are translocation assays developed for use with plant cells[13–15]. 'Nuclear translocation assay' (NTA), 'emerging circle of interactive proteins at specific endosomes' (ECLIPSE), and 'protein interactions from imaging of complexes after translocation' (PICT) are assays that require the addition of a compound (e.g., rapamycin) to monitor the translocation after the PPI[16–18]. Other translocation assays are based on oligomerization/aggregation readouts[19–21]

---

[1]Heidelberg University Biochemistry Center (BZH), Im Neuenheimer Feld 328, 69120 Heidelberg, Germany. ✉e-mail: jeske@bzh.uni-heidelberg.de

and may thus not be suitable for studying interactions with proteins that form granules on their own within a cell. Importantly, none of the translocation assays described have been evaluated for their ability to distinguish direct interactions from those in which the two proteins tested are potentially bridged by cell-endogenous proteins.

Here, we introduce a simple and rapid translocation PPI assay called ReLo for use with an animal cell culture. The assay is based on the relocalization of a protein upon its interaction with a second membrane-anchored protein. We apply ReLo to many large proteins, most of which have long disordered regions and are known to be insoluble after recombinant protein expression experiments. Using this set of proteins, we demonstrate that ReLo can be used to identify and characterize PPIs. Importantly, using two structurally well-characterized multidomain protein complexes, we show that ReLo detects only direct interactions in pairwise tests, a prerequisite for the analysis of previously unknown protein complexes by in vitro and structural biology methods. Using bridging ReLo experiments, we show that the binding topology of subunits within multiprotein complexes can be determined. We also use ReLo to identify protein domains that mediate complex formation, to test interfering point mutations, and to study interactions that depend on conformation or protein arginine methylation. In addition, ReLo is responsive to drug treatment, allowing the study of drug-induced interactions and the screening of small PPI inhibitors. In summary, ReLo is a simple, rapid, and versatile tool that allows the identification and thorough initial characterization of direct PPIs and PPI networks.

## Results

### ReLo: a simple and robust cell culture-based PPI assay

In preparation for the ReLo assay, two proteins of interest were fused to a red fluorescent protein (mCherry) and a green fluorescent protein (EGPF, mEGFP). Importantly, one of the constructs carried an additional fusion to a membrane-anchoring protein domain, resulting in a distinct subcellular membrane localization of the fusion protein. Upon interaction, the second protein is expected to colocalize with the anchored protein on the membrane and it would thereby relocalize with respect to its original location (Fig. 1a). Thus, we refer to the assay as the 'relocalization PPI assay', abbreviated 'ReLo'.

ReLo is based on a simple methodology in which cells are seeded on 4-well chamber coverslips and cotransfected with the desired combination of plasmids. After 48 h, the protein localization is analyzed by live-cell confocal fluorescence microscopy (Fig. 1a). We used S2R+ cells derived from semi-adherent Schneider's-line-2 (S2) cells, which were established from late *Drosophila* embryos[22,23]. Compared to S2 cells, S2R+ cells show greater adherence to dishes, and therefore do not require coated dishes for their adhesion prior to live-cell microscopy. All ReLo plasmids carry an in-frame blunt-end restriction site, allowing for a simple, fast, and straightforward cloning procedure (see "Methods" section).

To anchor cytoplasmic proteins of interest to a membrane, we selected the pleckstrin homology (PH) domain of the rat phospholipase C$\delta_1$ (PLC$\delta_1$), which specifically recognizes phosphatidylinositol 4,5-bisphosphate[24,25] and thus directs the fusion construct to the plasma membrane of a cell (Fig. 1a). In the ReLo assay, the membrane localization was independent of whether the PH domain was fused to the N- or C-terminus of a protein (Supplementary Fig. 1a). Unfortunately, nuclear proteins fused to the PH domain were only inefficiently retained in the cytoplasm and hardly localized to the plasma membrane (Supplementary Fig. 1c). Thus, testing of PPIs against a nuclear protein may lead to false-negative results if the interaction partner is located in the cytoplasm. Therefore, we tested alternative membrane-anchoring domains to assess their ability to retain nuclear proteins in the cytoplasm. We selected the mini membrane protein subunit 4 of the yeast oligosaccharyltransferase complex (OST4), which has previously been used as an N-terminal fusion protein to localize a nuclear

protein to the endoplasmic reticulum (ER)[26,27] (Supplementary Fig. 1d). As the plasma membrane localization of the PH domain is more distinct from a ubiquitous cytoplasmic localization than the ER localization of OST4, we prefer to use the PH domain in ReLo whenever possible.

### PPI mapping and mutational analysis

As a proof of concept for interaction studies with ReLo, we tested a previously characterized protein complex consisting of the extended LOTUS (eLOTUS) domain of Oskar (Oskar 139-240) and the C-terminal RecA-like domain (CTD) of the ATP-dependent DEAD-box RNA helicase Vasa (Vasa-CTD, Vasa 463-661)[28] (Fig. 1b). As only the short isoform of Oskar (Short Oskar, aa 139-606) interacts with Vasa[29], only Short Oskar and its domains were used in the following experiments (Fig. 1b). In our setup, the eLOTUS domain of Oskar was fused to PH-mCherry and localized to the plasma membrane. Vasa-CTD was fused to EGFP and localized ubiquitously in the cytoplasm and nucleus. When coexpressed with PH-mCherry-eLOTUS, but not with PH-mCherry alone, EGFP-Vasa-CTD relocalized to the plasma membrane (Fig. 1c). The unstructured region of Vasa (Vasa 1-200) and the N-terminal RecA-like domain (Vasa 200-463) did not interact with the Oskar-eLOTUS domain, and similarly, neither the unstructured region of Oskar (Oskar 241-387) nor its OSK domain (Oskar 388-606) interacted with Vasa-CTD (Fig. 1c). Surface point mutations, previously shown to interfere with the Vasa-Oskar interaction[28], were also found to be inhibitory in the ReLo assay (Fig. 1d). Together, these data confirmed the specific interaction between Vasa-CTD and Oskar-eLOTUS and demonstrated that ReLo can be used to map PPIs and to identify mutations that interfere with PPIs.

Oskar-eLOTUS and Vasa-CTD form a transient complex characterized by a dissociation constant ($K_D$) of ~10 μM, and although this complex has been crystallized, it is not stable enough to be detected by size exclusion chromatography[28,30,31]. Nevertheless, the relocalization upon the interaction between Oskar-eLOTUS and Vasa-CTD was clearly detectable in the ReLo assay: in a total of three independent replicates, the relocalization of Vasa-CTD towards the plasma membrane was observed in 91 out of 94 (i.e., 97%) cotransfected cells with both red and green fluorescence signals (Supplementary Fig. 2). These data indicate that relocalization is a very common event and that the assay is well suited for the study of low-affinity complexes. Most of the other interactions tested showed relocalization in 100% of the cotransfected cells (see below; Supplementary Fig. 3).

### Conformation-dependent interactions

Previous data suggest that the interaction between Oskar and Vasa depends on the conformation of Vasa[28]. We aimed to test the interaction between Oskar and different Vasa conformations in the context of a full-length protein. However, in contrast to the eLOTUS domain of Oskar, full-length Short Oskar was not localized to the cytoplasm but exclusively in the nucleus in S2R+ cells (Supplementary Fig. 1b)[28]. Therefore, we used the OST4-mCherry-Oskar construct, which localizes to membranous structures in the cytoplasm (Supplementary Fig. 1d, e). Upon coexpression, wild-type Vasa relocalized and colocalized with OST4-Oskar at the ER (Fig. 1e), confirming the interaction between Vasa and Oskar in the ReLo assay. Also using the OST4 anchor, the interaction mapped to the Oskar-eLOTUS domain and the Vasa-CTD (Supplementary Fig. 1e), and the known Vasa interface mutant (F504E) was unable to bind to Oskar (Fig. 1e), which is consistent with the data obtained using the PH domain (Fig. 1c, d).

The cores of Vasa and other ATP-dependent DEAD-box RNA helicases are composed of two RecA-like domains that adopt different orientations relative to each other depending on the presence of bound ATP and RNA[32]. In a substrate-unbound form, the helicase core adopts an open conformation and closes upon substrate binding (Fig. 2a). To assess the conformation-dependent Vasa interaction

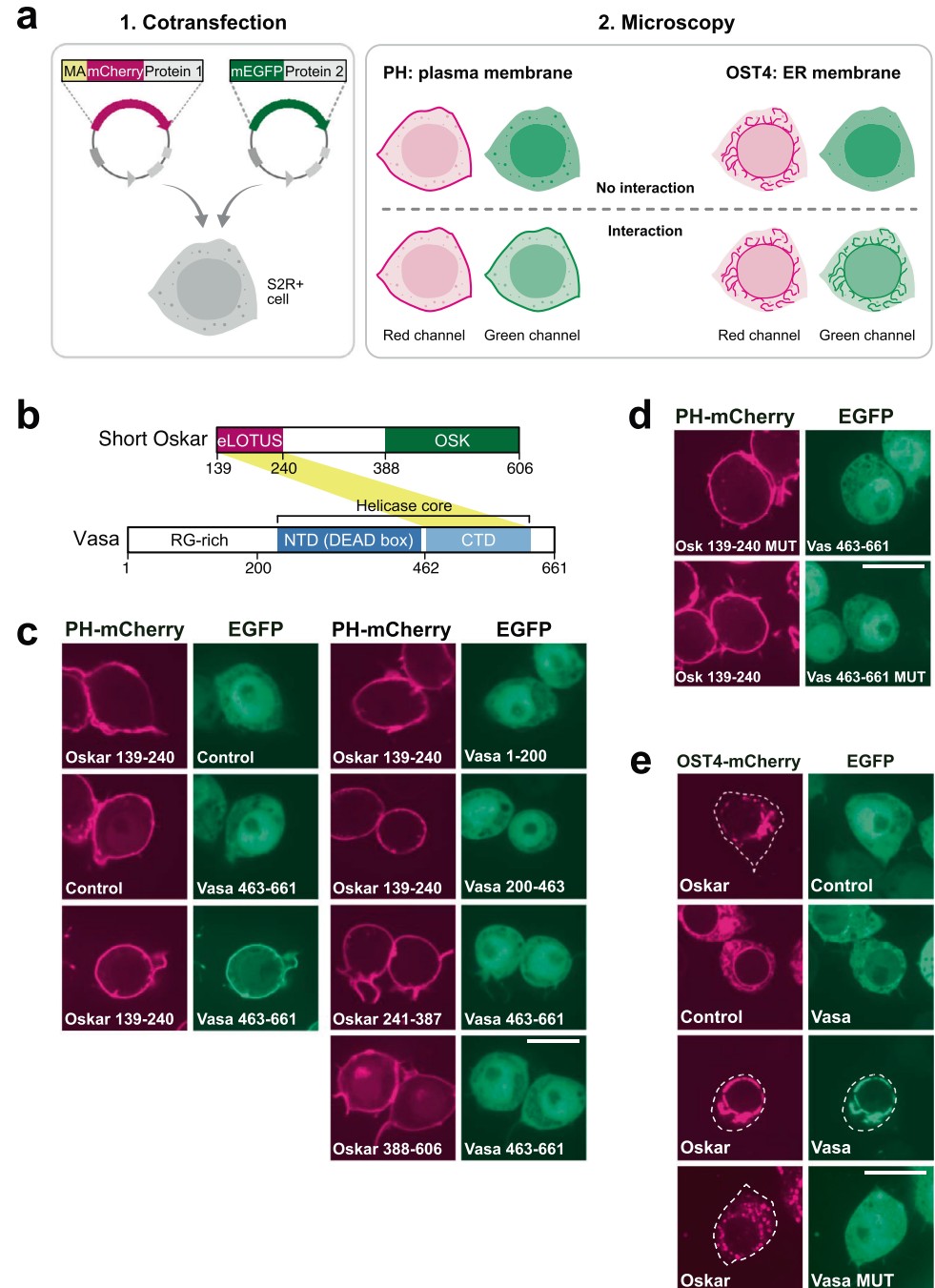

**Fig. 1 | ReLo assay and its use in PPI mapping and mutational analysis.**
**a** Plasmids encoding fluorescently tagged proteins 1 and 2 are cotransfected into
S2R+ cells. Protein 1 carries an additional fusion to a membrane-anchoring (MA)
domain, i.e., PH or OST4. After 48 h, protein localization is analyzed by confocal
fluorescence microscopy. If protein 2 interacts with protein 1, protein 2 is reloca-
lized to the plasma membrane (PH domain) or ER (OST4). The plasmid drawings
were created based on BioRender.com icons. **b** Domain organization of *Drosophila*
Oskar and Vasa proteins. Oskar is expressed in two isoforms, and Short Oskar lacks
amino acids 1-138. The eLOTUS domain of Oskar was previously shown to interact
with the C-terminal domain (CTD) of Vasa (yellow stripe), and the crystal structure

of the complex has been resolved[28,30]. **c** Vasa 463-661, but not Vasa 1-200 or 200-
463, interacted with the Oskar-eLOTUS domain. Vasa 463-661 did not interact with
Oskar 241-387 or Oskar 388-606. *n* = 5 experiments (left panel) and *n* = 1 experiment
(right panel), of which the data are consistent with the experiment shown in Sup-
plementary Fig. 1e. **d** Oskar A162E/L228E and Vasa F504E point mutations (MUT)
interfered with the Oskar-Vasa interaction. *n* = 2 experiments. **e** OST4-mCherry
Oskar localized to the ER (top panel), and Vasa relocalized with Oskar to the ER
(lower mid panel), while Vasa MUT did not (bottom panel). *n* = 2 experiments. In the
"Control" experiments the respective construct was coexpressed with the vector
indicated at the top of the panel lacking an insert. The scale bar is 10 μm.

with the ReLo assay, we used Vasa variants with well-characterized
point mutations in the ATP-binding pocket that stabilize either the
open conformation (K295N; Vasa-open) or the closed conformation
(E400Q; Vasa-closed)[33–35]. Neither mutation is located in the Vasa-
Oskar binding interface. When we tested the interaction of OST4-
anchored Oskar with the Vasa mutants using the ReLo assay, we

observed an interaction with Vasa-open but not Vasa-closed (Fig. 2b),
indicating that Oskar preferentially binds to the open conformation
of Vasa. These results are consistent with our previous
observations[28], and demonstrate that the ReLo assay allows the study
of PPIs that depend on the specific conformation of an interaction
partner.

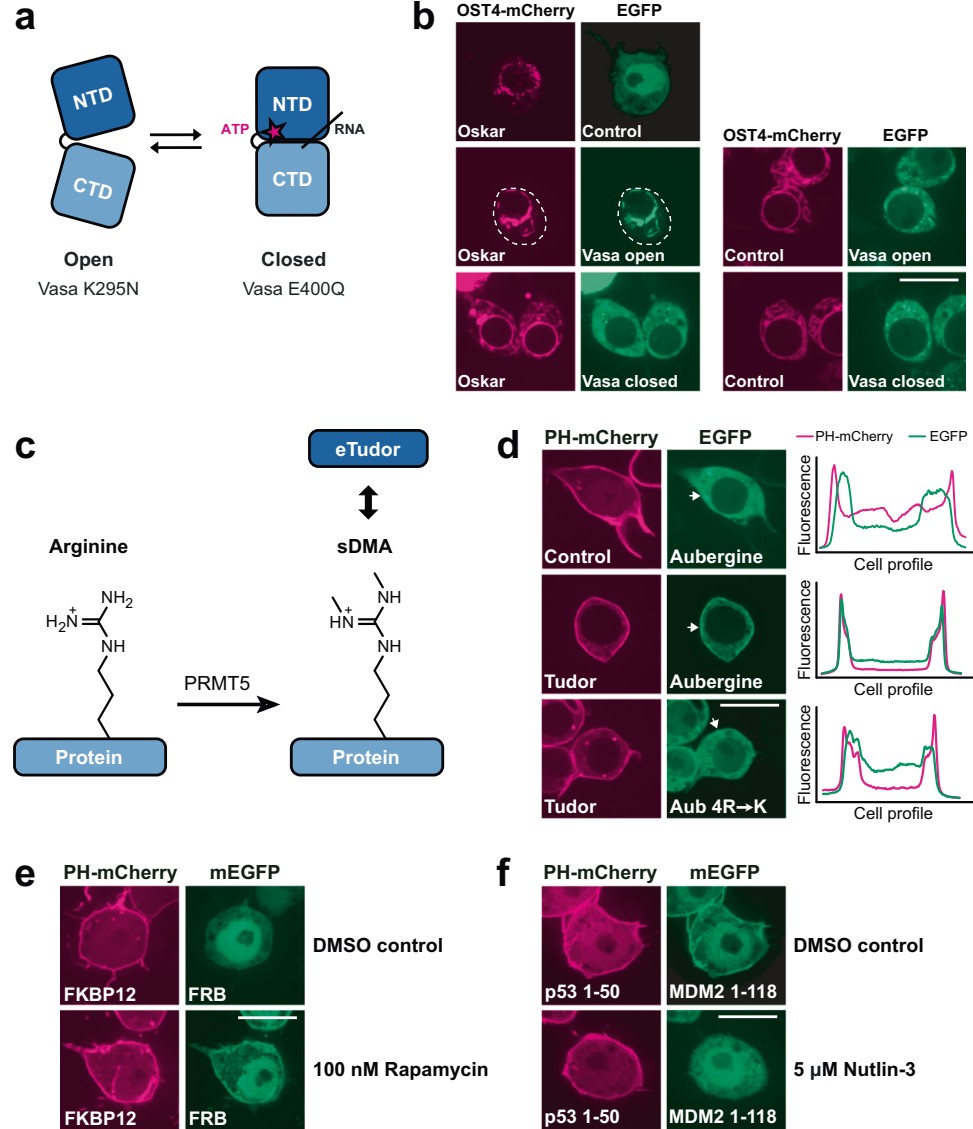

**Fig. 2 | PPI studies using ReLo with respect to conformation, post-translational modification, and drug sensitivity. a** Pictorial representation of Vasa N- and C-terminal RecA-like domains (NTD and CTD, respectively) in the open and closed conformations. Vasa-open and Vasa-closed carry the K295N or E400Q mutations, respectively. **b** EGFP-Vasa-open, but not EGFP-Vasa-closed, interacted with OST4-mCherry-Oskar. The white dotted lines indicate cell boundaries. *n* = 2 experiments. **c** Symmetric dimethyl arginine (sDMA) modifications can be catalyzed by PRMT5 and recognized by eTud domains. **d** Wild-type Aub interacted with Tudor, whereas Aub carrying nonmethylatable 4R→K point mutations (R11K/R13K/R15K/R17K) did

not. The profiles indicate the measured fluorescence distribution across the cell along a line indicated by the white arrows (shown for the green channel only). Red and green fluorescence signals are independent and displayed relatively. *n* = 3 experiments. **e** Rapamycin induced an interaction between human FKBP12 and FRB. The control contained 0.0009% dimethyl sulfoxide (DMSO). *n* = 2 experiments. **f** The interaction between p53 1–50 and MDM2 1-118 was inhibited by nutlin-3 treatment. The control contained 0.05% DMSO. *n* = 2 experiments. In the "Control" experiments the respective construct was coexpressed with the vector indicated at the top of the panel lacking an insert. The scale bar is 10 µm.

## Protein arginine methylation-dependent interactions

Protein interactions may depend on post-translational modifications, such as protein arginine methylation. The symmetric dimethylated arginine (sDMA) modification is catalyzed by a subset of protein arginine methyltransferases (PRMTs)[36], and sDMA methylation activity has been previously reported in S2 cells[37]. To test whether the ReLo assay is suitable for investigating interactions involving sDMA modifications in S2R+ cells, we tested the previously characterized strictly sDMA-dependent interaction between the PIWI protein Aubergine (Aub) and Tudor[38,39] (Fig. 2c). Using ReLo, we indeed observed an Aub-Tudor interaction (Fig. 2d). Aub carries four sDMAs within its RG-rich N-terminus (R11, R13, R15, and R17), which are specifically recognized and bound by extended Tudor (eTud) domains of Tudor[38–41]. Substitution of these four arginine residues with lysine residues (Aub R→K)

rendered Aub unmodifiable by PRMT5[41] and abolished the Aub-Tudor interaction (Fig. 2d). Taken together, these data demonstrate that the S2R+ cells have sufficient sDMA activity to effectively modify proteins expressed after transient transfection of the cells. We conclude that ReLo is suitable for the study of PPIs that depend on PRMT5-catalyzed sDMA modification.

## Effect of small molecules on PPIs

Next, we tested whether the ReLo assay could be used to study PPIs that are induced by the addition of small molecules to the cell culture medium. To this end, we tested the previously characterized rapamycin-dependent interaction between human FK506-binding protein 12 (FKBP12) and the FKBP12 rapamycin binding domain (FRB) of human mTOR[42]. In the absence of rapamycin, using the

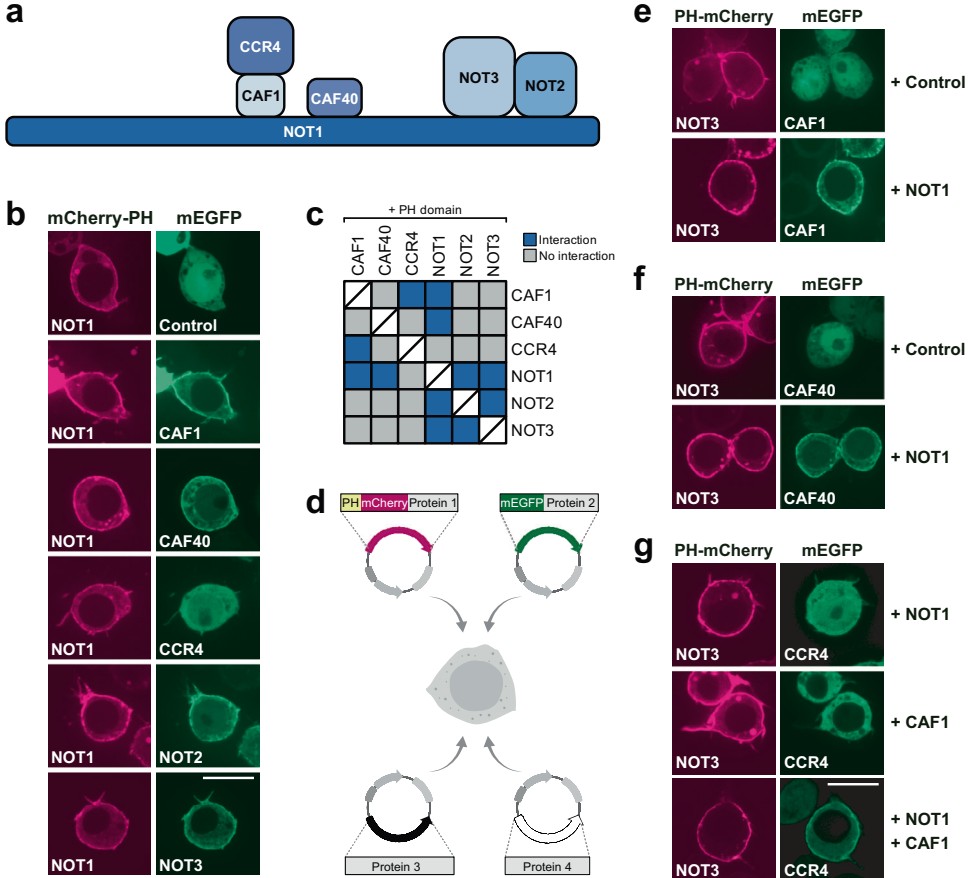

**Fig. 3 | ReLo identifies direct PPIs. a** Subunit organization of the *Drosophila* CCR4-NOT core complex. **b** NOT1-mCherry-PH recruited CAF1, CAF40, NOT2, and NOT3 but not CCR4 to the plasma membrane. See Supplementary Fig. 3a for the PPI analysis of the other CCR4-NOT complex subunits. *n* = 1 experiment, of which the data are fully consistent with the experiments using swapped tags shown in Supplementary Fig. 3a. **c** Summary of the results from a pairwise screen of CCR4-NOT complex core subunits using ReLo. The blue color indicates an interaction, and the gray color indicates no interaction. **d** Schematic showing the ReLo assay involving coexpression of four different protein constructs. The plasmid drawings were created based on BioRender.com icons. **e** NOT3 interacted with CAF1 in the presence of NOT1. *n* = 2 experiments. **f** NOT3 interacted with CAF40 in the presence of NOT1. *n* = 3 experiments. **g** NOT3 recruited CCR4 to the plasma membrane only in the presence of both NOT1 and CAF1. *n* = 2 experiments. In the "Control" experiments the respective construct was coexpressed with the vector indicated at the top of the panel lacking an insert. The scale bar is 10 μm.

dimethyl sulfoxide (DMSO)-containing control medium, FKBP12, and FRB did not interact in the ReLo assay but did interact in the presence of 100 nM rapamycin in the cell culture medium (Fig. 2e).

We also tested whether the ReLo assay can be used to inhibit PPIs by drug treatment. We chose to interfere with the interaction between human p53 and the human ortholog of mouse double minute 2 (MDM2) using the known peptidomimetic inhibitor nutlin-3[43]. In the ReLo assay, we used only the N-terminal domains of p53 and MDM2, which are sufficient to mediate the p53-MDM2 interaction[44]. Indeed, although an interaction between p53 1–50 and MDM2 1-118 was detected in the DMSO control experiment, this interaction was not observed when the cells were incubated with 5 μM nutlin-3 (Fig. 2f).

Taken together, these data suggest that the ReLo assay can be used to study interactions with non-*Drosophila* proteins and to screen drugs for their ability to either enable or inhibit a specific PPI.

**ReLo reveals direct interactions**

So far, we have tested known direct interactions between proteins that are not endogenously expressed in S2R+ cells. PPIs detected by the ReLo assay do not necessarily involve direct contact. Instead, PPIs may also result from indirect contacts caused by incorporation of the two coexpressed proteins into cell-endogenous protein complexes and subsequent indirect bridging of this pair by one or more common interaction partners. To understand the extent to which direct or indirect associations underlie a relocalization event observed with ReLo, we examined interactions between individual subunits of the CCR4-NOT complex, which is endogenous to S2R+ cells. The CCR4-NOT complex is an essential eukaryotic deadenylase composed of six subunits that form the core with an architecture that is well characterized at the molecular and structural levels[45,46]. In this complex, NOT1, which has a size of 281 kDa in *Drosophila*, acts as a scaffolding subunit for the assembly of all other subunits. Specifically, the CAF1-CCR4 heterodimer and CAF40 bind to the central region of NOT1, and the NOT2-NOT3 subcomplex associates with the C-terminal region of NOT1 (Fig. 3a). Using ReLo, we performed a systematic pairwise screen testing each subunit against all other subunits of the *Drosophila* CCR4-NOT core complex (Fig. 3b and Supplementary Fig. 4a). In S2R+ cells, NOT1 and NOT3 localized exclusively to the cytoplasm, whereas NOT2, CAF1, CAF40, and CCR4 localized to both the cyto- and the nucleoplasm (Supplementary Fig. 4b). Notably, we observed interactions only between proteins that had previously been shown to exhibit direct associations, but not between indirectly linked combinations: NOT1 bound specifically to CAF1, CAF40, NOT2, and NOT3, whereas CAF1 bound to CCR4, and NOT2 bound to NOT3 (Fig. 3b, c, Supplementary Fig. 4a, b). We also tested the interactions to CAF1 using the OST4 membrane anchor and observed similar results to those obtained using the PH anchor (Supplementary Fig. 4c). For NOT2 and NOT3, two example proteins for which we had antibodies available, we compared

the endogenous expression levels with those of the respective transfected constructs in four and three replicates, respectively, and observed that the degree of overexpression varied between 2- and 27-fold (Supplementary Fig. 5). Taken together, these data suggest that, probably due to overexpression, the incorporation of two tagged CCR4-NOT complex subunits into an endogenous complex is insignificant, and thus direct but not indirect interactions between the two proteins tested were observed with the ReLo assay.

Although the subunits of the endogenous CCR4-NOT complex show moderate to high expression levels in S2R+ cells[47] (Supplementary Fig. 6a), we wanted to challenge the ReLo assay by testing PPI between subunits of a cytoplasmic complex that is even more abundant in S2R+ cells. We chose the seven-subunit Arp2/3 complex, which is known to initiate actin polymerization in eukaryotes[48] and of which several subunits are among the most highly expressed cytoskeletal genes[47,49] (Supplementary Fig. 6a). In S2R+ cells, the Arp2/3 subunits Arp3, Arpc1, and Arpc4 are expressed approximately two- to four-fold higher as compared to NOT3, which is the most abundant subunit of the CCR4-NOT complex (Supplementary Fig. 6a). Compared to the CCR4-NOT complex, the binary PPIs within the Arp2/3 complex are less well characterized, and we used the crystal structure of the bovine Arp2/3 complex[50] to infer which subunit is directly or indirectly bound to another subunit (Supplementary Fig. 6b, c). The crystal structure suggests six direct pairwise contacts. Using ReLo assays, we detected two PPIs, namely the Arpc2-Arpc4 and the Arpc4-Arpc5 interaction (Supplementary Fig. 6c, d). The other four contacts that we did not observe with ReLo, were also not detected in a previous pairwise study of the human Arp2/3 complex using Y2H assays[51]. Importantly, we did not detect any false-positive interactions using the ReLo assay. These data suggest that even when a complex is highly abundant in a cell, the detection of a PPI between two protein candidates in the ReLo assays is direct and not indirectly mediated by incorporation into the same endogenous complex. Nevertheless, we cannot exclude the possibility that indirect interactions via a naturally expressed bridging protein may occasionally be detected.

Finally, we compared the ReLo assay with the split-ubiquitin-based membrane yeast two-hybrid (MYTH) assay, in which, unlike the classical Y2H assay, the interaction occurs in the cytoplasm[27,52]. We have previously used MYTH successfully to map the interaction between Short Oskar (54 kDa) and Vasa (72 kDa)[30]. When we assessed the pairwise interactions between the six core subunits of the CCR4-NOT complex (34–281 kDa) using MYTH, we observed mostly false-negative and false-positive results; the only conclusive interaction that we detected was between NOT2 and NOT3 (Supplementary Fig. 7). These data indicate that, although MYTH and ReLo worked well to map the Oskar-Vasa interaction, ReLo might be expected to be more reliable than yeast-based assays for testing (large) proteins from higher eukaryotes, although a larger scale study would be required to fully test this.

## Topological description of multisubunit complexes

As has been described for S2 cells[53], we usually observed a very high cotransfection efficiency of the S2R+ cells when performing ReLo experiments. Therefore, we tested whether the bridging of CCR4-NOT subunits that do not directly interact can be observed when one or two common binding partners are added to the mixture. Specifically, S2R+ cells were cotransfected with three or four plasmids: one plasmid expressing the PH-mCherry fusion construct, one plasmid expressing the mEGFP fusion construct, and one or two plasmids expressing non-fluorescent bridging factors (Fig. 3d). Indeed, NOT3 did not interact with CAF1 or CAF40 in the presence of the control plasmid but interacted when NOT1 was added (Fig. 3e, f). We also tested the bridging of the NOT3-CCR4 interaction, which requires not only NOT1 but also the CAF1 subunit. The NOT3-CCR4 interaction was not observed when NOT1 alone was coexpressed but was observed when both NOT1 and

CAF1 were coexpressed (Fig. 3g). Similar results were obtained when testing for indirect interactions with CAF40 (Supplementary Fig. 4d). Taken together, these data demonstrate that bridging experiments with ReLo can be used to reconstruct the binding topology of multi-subunit complexes such as the CCR4-NOT complex.

## Evaluation of PPIs between the CCR4-NOT complex and mRNA repressor proteins

The CCR4-NOT complex can be specifically recruited to mRNAs by adapter RNA-binding proteins, leading to accelerated deadenylation and degradation and/or translational repression of the targeted mRNA. We examined the PPIs between some of these repressor proteins and the six core subunits of the CCR4-NOT complex using ReLo (Fig. 4a). Co-IP experiments combined with structural analysis have previously revealed the specific subunit(s) of the CCR4-NOT complex through which adapter proteins recruit the complex to a specific mRNA. For example, *Drosophila* Bag-of-marbles (Bam) was shown to specifically bind to CAF40[54]. We confirmed this finding with ReLo: Bam bound to CAF40, but not to any other subunit of the CCR4-NOT complex (Fig. 4b and Supplementary Fig. 8a). A point mutation in Bam (M24E; Bam MUT), which is known to interfere with CAF40 binding[54], also abolished CAF40 binding in the ReLo assay (Fig. 4b). Similarly, a mutation on the CAF40 surface (V186E; CAF40 MUT), which has been shown to prevent Bam binding[54], abolished the interaction with Bam in the ReLo assay (Fig. 4b).

*Drosophila* Roquin has been shown to bind to *Drosophila* CAF40. It also binds to the so-called CAF40 module, which consists of CAF40 and the CAF40-binding region of NOT1, and to the NOT1/2/3 module, which consists of the C-terminal domains of the NOT1, NOT2, and NOT3 subunits of the human CCR4-NOT complex[55]. Among the subunits tested in the ReLo assay, we detected clear binding of Roquin to *Drosophila* CAF40, NOT1, and NOT3 but not to NOT2 (Fig. 4c and Supplementary Fig. 8b), essentially confirming the previous data. In addition, we detected a previously unknown interaction between Roquin and CCR4 (Fig. 4c).

*Drosophila* Nanos has been shown to bind to the NOT1/2/3 module of the human CCR4-NOT complex in GST pull-down experiments but does not bind to any of these domains individually[56]. Using ReLo, we detected the interaction of Nanos with *Drosophila* NOT1 but not with NOT2 or NOT3, suggesting that the NOT1 interaction may be predominant (Fig. 4d and Supplementary Fig. 8c). In addition, we detected Nanos binding to the CAF40 subunit (Fig. 4d), an interaction that has not been previously reported.

Encouraged by these largely confirmatory data, we tested the interaction of the CCR4-NOT complex with less well-characterized potential adapter proteins. *Drosophila* meiosis regulator and mRNA stability factor 1 (MARF1) is an oocyte-specific protein that recruits the CCR4-NOT complex to target mRNAs, thereby controlling meiosis[57]. *Drosophila* Smaug recruits the CCR4-NOT complex to *nanos* and other mRNAs to regulate posterior patterning and nuclear divisions of the early embryo[58–62]. Whether MARF1 or Smaug directly associates with the CCR4-NOT complex was unknown. Using ReLo, we found that Smaug interacts with the NOT3 subunit[63] and that MARF1 binds to the NOT1 subunit (Fig. 4e and Supplementary Fig. 8d), suggesting that Smaug and MARF1 are indeed direct recruiters of the CCR4-NOT complex.

*Drosophila* Cup and its ortholog 4E-T are translation repressor proteins that act through their interactions with the eukaryotic translation initiation factor 4E (eIF4E) and the DEAD-box RNA helicase Me31B/DDX6[64–68]. Both the human 4E-T - Me31B and the *Drosophila* Cup-eIF4E complex have been structurally characterized[69,70], and we detected both interactions using ReLo assays (Fig. 4f, left panels). As shown by co-IP experiments, Cup also interacts with the CAF1, CCR4, NOT1, NOT2, and NOT3 subunits of the CCR4-NOT complex[71]. We did not detect Cup binding to any subunit of the CCR4-NOT core complex

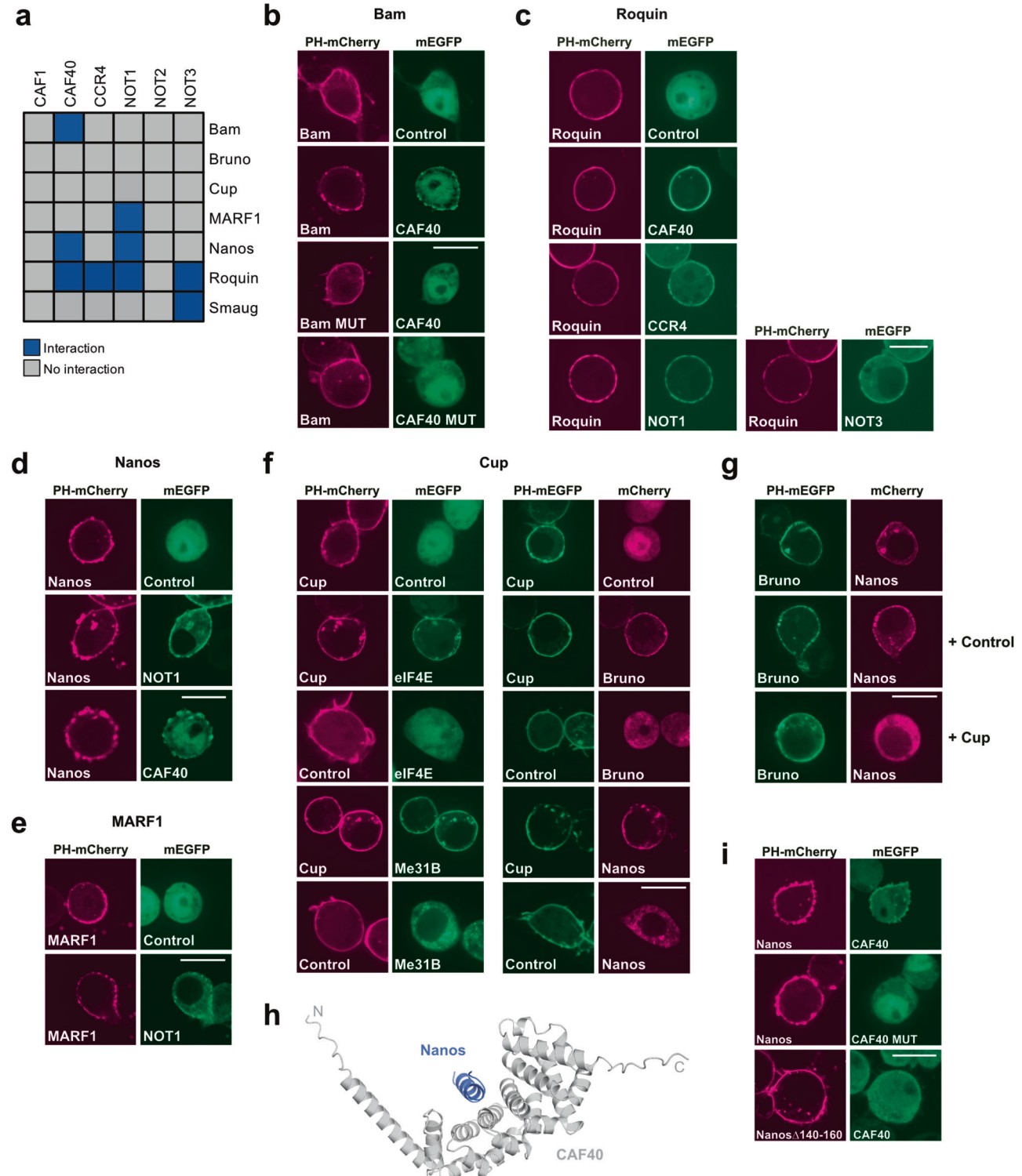

**Fig. 4 | PPIs between the CCR4-NOT complex and repressor proteins.**
**a** Summary of the results from a pairwise screen of CCR4-NOT complex core subunits with RNA-binding proteins as indicated. The blue color indicates an interaction, and the gray color indicates no interaction. Interaction data for Bruno and Cup are shown in Supplementary Fig. 7 and data for Smaug are published in[63]. **b** Bam interacted with CAF40. Bam MUT carried the M24E and CAF40 MUT the V186E point mutation. **c** Roquin interacted with CAF40, CCR4, NOT1, and NOT3. $n = 2$ experiments. **d** Nanos interacted with NOT1 and CAF40. **e** MARF1 interacted

with NOT1. **f** Cup interacted with eIF4E, Me31B, Bruno, and Nanos. **g** Bruno interacted with Nanos in the absence but not in the presence of Cup. **h** Structural model of the CAF40-Nanos 140–160 complex obtained using AlphaFold-Multimer version 3. The predicted aligned error (PAE) plot is shown in Supplementary Fig. 9. **i** CAF40 MUT (V186E) did not interact with Nanos and Nanos lacking residues 140-160 did not interact with CAF40. In the "Control" experiments the respective construct was coexpressed with the vector indicated at the top of the panel lacking an insert. For all, $n = 2$ experiments. The scale bar is 10 μm.

using ReLo assays (Supplementary Fig. 8e), suggesting that if Cup directly recruits the CCR4-NOT complex, it is likely to involve weak and multivalent interactions. Cup has also been reported to bind the repressors Nanos and Bruno[65,72,73], and we confirmed both interactions using the ReLo assay (Fig. 4f, right panels). Consistent with a recent report[74], we also observed that Bruno directly binds to Nanos (Fig. 4g). We then asked whether Bruno and Nanos bind to Cup as a complex. However, when coexpressed with Cup, Bruno and Nanos did not interact (Fig. 4g), suggesting that Bruno and Nanos compete for binding to the same region of Cup. Bruno has thus far not been shown to directly contact the CCR4-NOT complex, and we did not detect Bruno binding to any of the core subunits of the CCR4-NOT complex (Supplementary Fig. 8f).

In conclusion, the evaluation of PPIs using ReLo with a broad selection of repressor proteins confirmed many previously described interactions and revealed some that, to the best of our knowledge, were not previously described. Thus, ReLo can be used to identify PPIs in a candidate approach.

### Combining ReLo assays with structural prediction

Finally, we wanted to gain some molecular insight into one of the interactions that had not been described before. We tested the structural prediction of the Nanos-CAF40 complex using AlphaFold-multimer-v3 and obtained a high-confidence model as indicated by the predicted aligned error (PAE) plot (Fig. 4h, Supplementary Fig. 9a)[75–77]. The predicted structure is composed of full-length CAF40 and a short α-helix spanning residues 140–160 of Nanos, which binds to the same site of CAF40 that has been shown to accommodate the Bam or Roquin α-helical peptide[54,55] (Supplementary Fig. 9b). The CAF40 point mutation, which prevented Bam binding (Fig. 4b), also abolished the interaction with Nanos (Fig. 4i). Similarly, deletion of the CAF40-binding α-helix from Nanos (Nanos Δ140–160) prevented the interaction with CAF40 (Fig. 4i). These data are consistent with the predicted structural model of the Nanos-CAF40 complex. Taken together, our exemplary analysis suggests that the combination of ReLo assays with structural modeling can lead to rapid molecular insight into protein complexes.

## Discussion

Detection and characterization of PPIs are critical for uncovering regulatory mechanisms that underlie cellular processes. Here, we describe ReLo, a rapid method for the identification and study of pairwise and multisubunit PPIs. We provide evidence showing that pairwise PPIs identified by ReLo are based on direct contacts. Thus, PPI partners identified by ReLo are promising candidates for use in subsequent studies using in vitro assays and experimental structural biology methods. Alternatively, PPI mapping data obtained from ReLo experiments can be used to guide subsequent modeling experiments using AlphaFold-Multimer to obtain structural information on complexes[75,76]. The predicted protein–protein interface can then be rapidly validated by mutational analysis using ReLo, and the mutations that specifically disrupt a PPI can eventually be tested in vivo without the need to purify a protein or experimentally determine the structure of a protein complex.

We refrained from quantifying the relocalization events observed with ReLo using image processing tools to evaluate or compare PPIs. Although only qualitative, the results obtained were typically unambiguous. For example, in control experiments, we never observed relocalization, and in experiments where proteins interacted, most or all cells showed relocalization (Supplementary Figs. 2 and 3). Furthermore, implementing a relocalization score as a quantified readout of an interaction may have easily led to the false assumption that the score represents a measure of binding affinity between proteins. Such false assumptions are also a problem with Y2H dot assays, which appear to provide quantitative data but do not. The degree of

relocalization identified with the ReLo assay or yeast cell growth in a Y2H assay does not only reflect the binding affinity of the protein complex, but also depends on the protein expression levels, protein stability, subcellular localization, and other factors. With this thought in mind, we prefer to consider ReLo as a qualitative PPI method whose strength lies in its speed and versatility.

ReLo is easy to perform in laboratories equipped with cell culture technology and with access to a confocal fluorescence microscope. Unlike standard Y2H or PCA assays, the expression levels of the proteins tested in a ReLo assay are monitored directly during microscopy, which facilitates data interpretation. The *Drosophila* S2R+ cell line used for our ReLo tests is easy to handle. Similar to S2 cells, S2R+ cells grow at room temperature, do not require an incubator with $CO_2$, and can be passaged without the need for coated dishes or scraping or trypsinization of cells[53]. Using S2R+ cells, we have successfully investigated not only *Drosophila* PPIs but also human PPIs. However, if a cell line derived from an alternative organism is required for a ReLo assay, only the PH or OST4 membrane anchors need to be inserted into expression vectors compatible with the desired cell line. For proteins with strong intrinsic localization signals, it may be necessary to remove such signals before testing for an interaction-induced relocalization.

Using ReLo assays, we investigated PPIs involving NOT1 and Tudor, two large proteins of 281 kDa (2505 aa) and 285 kDa (2515 aa), respectively. Thus, the detection of PPIs using the ReLo assay appears to be successful regardless of the length of the protein of interest, which, in our experience, is a major advantage over the yeast split-ubiquitin system. As with other cell-based PPI assays, care must be taken when working with toxic proteins. To reduce toxicity, we recommend testing the splitting of the toxic protein into its domains or, if possible, testing protein variants with mutated active sites. Supplementary Table 1 provides a comparison of the ReLo assay with other common cell-based PPI methods.

PPIs are highly relevant as putative therapeutic targets for the development of new treatments[78,79]. In ReLo, complex formation is reversible, and we have demonstrated that ReLo is a tool for testing the effect of small molecules on PPIs. Because of its simple setup, ReLo could be adapted to a high-throughput approach using automated imaging, allowing for large drug screening experiments. In a setup where a single specific PPI is subjected to a drug screening experiment, it may be advantageous to express the two protein partners from one plasmid.

Taken together, our data show that ReLo assays are fast, simple, and robust. We recommend using ReLo as an initial tool to screen and characterize PPIs, especially in cases where yeast-based methods or more complicated methods fail. Subsequently, ReLo can be complemented with biochemical, structural, or genetic approaches to further characterize or ultimately validate the biological relevance of a given PPI.

## Methods

### Plasmid backbone construction

pAc5.1-EGFP and pAc5.1-mCherry were previously described[28]. pAc5.1-mEGFP encodes the monomeric A206K mutant EGFP variant and was generated by site-directed mutagenesis of pAc5.1-EGFP. The pAc5.1-λN-HA vector[80] was used to generate non-fluorescent constructs. The *Saccharomyces cerevisiae* OST4 sequence was amplified from the pDHB1 vector[27] and inserted into the KpnI site of pAc5.1-mCherry to yield pAc5.1-OST4-mCherry. For cloning into the pAc5.1-EGFP, pAc5.1-mEGFP, pAc5.1-mCherry, pAc5.1-λN-HA, and pAc5.1-OST4-mCherry vectors, sequences of interest were inserted into the blunt-end EcoRV site. The rat PLCδ₁-PH sequence was amplified from the pETM11-His6-PH-Sumo3-sfGFP vector[81] and inserted into the KpnI site of pAc5.1-mCherry and pAc5.1-mEGFP to yield pAc5.1-PH-mCherry and pAc5.1-PH-mEGFP, respectively. Alternatively, the PH sequence was inserted into the EcoRV site of pAc5.1-mCherry to obtain the pAc5.1-mCherry-

PH vector. For all PH-containing vectors, a unique in-frame FspAI blunt-end site was introduced 3′ or 5′ with respect to the fluorescent protein sequences and was used to insert the sequence of interest. The OST4 and PH sequences are provided in the Supplementary Information file.

## Cloning

Ligation reactions were assembled in a 10 μl reaction containing T4 DNA ligase (Thermo Scientific), 50 ng of vector DNA, and a 5- to 20-fold molar excess of DNA inserts and were incubated for 1–2 h at room temperature. DNA inserts were produced by PCR amplification. To prevent re-ligation, the reaction was supplemented with 0.25 μl of the blunt-end restriction enzyme that was also used for linearization of the respective vector, unless this site was present in the insert sequence. Positive clones were screened by colony PCR using one primer that binds to the vector and a second primer that binds to the insert. All constructs were verified by sequencing. Detailed information on all plasmids used in this study is provided in Supplementary Table 2. Primer sequences are provided as Supplementary Table 3.

## Cell culture

S2R+ cells were obtained from the lab of Aurelio Teleman and cultured at 25 °C in Schneider's *Drosophila* medium + (L)-glutamine (Thermo Scientific) supplemented with 10% fetal bovine serum (Sigma) and 1 × Gibco™ Antibiotic-Antimycotic (Thermo Scientific). Cells were seeded into six-well glass-bottom plates (Cellvis; earlier experiments) or four-well polymer μ-slides (Ibidi; later experiments) and cotransfected using jetOPTIMUS (Polyplus Transfection) according to the user manual. Specifically, 600 μl of cells at a density of $1 × 10^6$ cells/ml were seeded per well in a four-well slide and 61 μl of transfection reagent mixture was added. This mixture contained 1 μl of transfection reagent and a total of 600 ng of DNA for cotransfection of two (300 ng DNA each), three (200 ng DNA each), or four (150 ng DNA each) plasmids diluted in jetOPTIMUS buffer. After 48 h of incubation at 25 °C, images of live cells were captured with either a Nikon Plan Apoλ 100× NA 1.45 oil objective and a Perkin Elmer Ulraview VoX System using a Yokogawa X-1 spinning disk scanhead (Hamamatsu C9100-50 camera, 840 × 1000 resolution), a Nikon Plan Apoλ 100× NA 1.45 oil objective and Perkin Elmer Ulraview VoX System using a Yokogawa X-1 spinning disk scanhead (Hamamatsu C9100-02 EMCCD camera, 1000 × 1000 resolution), or a Nikon Apo 60× NA 1.40 oil-λS objective and a Nikon Eclipse Ti2 inverted microscope (Nikon confocal Ax camera, 1024 × 1024 resolution). The acquisitions software was Volocity for the first two microscopes and NIS Elements AR (version 5.41.02) for the latter. Images were processed with Fiji software (ImageJ version 1.53t)[82]. The average cotransfection efficiency was approximately 30%. Changing the amount of DNA or transfection reagent during the transfection did not significantly change the protein expression levels in the cells. However, reducing the total amount of DNA to 150 ng and/or the amount of transfection reagent to 0.5 μl significantly reduced the transfection efficiency.

## Drug treatment

Rapamycin and Nutlin-3 (both from MedChemExpress) were dissolved in DMSO to obtain stock concentrations of 10.9 mM and 10 mM, respectively, and they were diluted with serum-free SF-4 Baculo express medium (BioConcept) to obtain working concentrations of 100 nM and 5 μM, respectively. S2R+ cells were grown in SF-4 medium and cotransfected with the desired plasmids using FuGENE® HD transfection reagent (Promega). Specifically, cells were seeded as described above and 26 μl of transfection reagent mixture was added. This mixture contained 1 μl of transfection reagent and 400 ng of DNA (i.e., 200 ng DNA each for cotransfection of two plasmids) diluted in SF-4 Baculo express medium. Twenty-four hours after transfection, the cells were treated with the drug by replacing the medium with drug-containing medium. Cells were imaged after 24 h of incubation with the drug or DMSO control medium at 25 °C.

## Western blots

S2R+ cells were harvested by centrifugation at 13,000 × *g* for 3 min and lysed in 5 × SDS sample buffer (10% SDS, 250 mM Tris-HCl pH 6.8, 0.5% β-mercaptoethanol, 50% glycerol, 0.1% bromophenol blue). Samples were heated at 95 °C for 10 min and proteins were separated by SDS-polyacrylamide gel electrophoresis. Proteins were then transferred to a nitrocellulose membrane (GE Healthcare) using a semi-dry electro-blotting system (Thermo Scientific) at max. 14 V for 1 h in Towbin buffer (25 mM Tris-HCl, pH 7.5, 192 mM glycine, and 20% (v/v) methanol). After transfer, the membrane was blocked with 5% milk powder in 1 × TN-Tween buffer (20 mM Tris-HCl, pH 7.5, 150 mM NaCl, and 0.05% Tween 20) either overnight at 4 °C or for 1 h at RT. Incubation with anti-NOT2 (SA3859) or anti-NOT3 (SA4143) antibodies (both described in ref. [83]) in blocking solution at dilutions of 1:1000 and 1:333, respectively, was performed for 2 h at RT. Incubation with the peroxidase-conjugated donkey anti-rabbit secondary antibody (Cytiva, NA934, lot 17824033) at 1:10,000 dilution was performed for 1 h at RT. Protein bands were detected using ECL western blotting substrate (Thermo Scientific).

## Split-ubiquitin-based membrane yeast two-hybrid assay (MYTH)

MYTH assays were performed as described previously[30]. 500 ng each of bait pPR3-N and prey pDHB1 vectors were cotransformed into competent NMY51 yeast cells, which were then plated onto SDC agar lacking leucine and tryptophan and incubated at 30 °C for two days. To perform the spotting assay, three to five colonies were picked, resuspended in water, and the cell suspension was then diluted to an $OD_{600}$ of 3 and to four more consecutive 1:10 dilutions. 5 μl of each dilution was spotted on SDC agar plates either lacking leucine and tryptophan (control plates) or lacking leucine, tryptophan, adenine, and histidine (selection plate). The plates were then incubated for two (control plate) or six (selection plate) days at 30 °C and images were taken.

## Structural prediction

For the prediction of the structure of the Nanos-CAF40 complex the ColabFold v1.5.2 web interface[77] was used with standard settings except for the model_type, which was switched from "auto" to "alphaFold_multimer_v3". Structures were visualized using Pymol (version 2.4.0).

## Statistics & reproducibility

For ReLo experiments, one or two cells are shown per image, which are representative to the whole cotransfected cell population (three examples are shown in Supplementary Figs. 2 and 3). Transfection experiments were performed two or more times. We consider experiments with swapped or different tags also as replicates. The number of replicates is indicated in the figure legends. Negative control experiments were generally performed only once, because the localization of the construct containing the insert was consistent between these negative controls and other negative results, in which the construct was tested for interaction with another protein. No statistical method was used to predetermine sample size. Cells that either contained a nucleus with an unusual shape, that did not contain a compact nucleolus, or that contained gigantic vesicles were excluded from the analyses. The experiments were not randomized. The investigators were not blinded to allocation during experiments and outcome assessment.

## Reporting summary

Further information on research design is available in the Nature Portfolio Reporting Summary linked to this article.

## Data availability

The authors declare that all relevant data supporting the findings of this study are available within the article and its supplementary information files. The crystal structures used in this study are available in the PDB database under accession codes 1K8K, 5ONB, and 5LSW. Uncropped blots are provided with this paper at the end of the Supplementary information file.

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

## Acknowledgements

We thank Elmar Wahle, Julien Bethune, Bernd Bukau, and Ivana Vonkova for DNA constructs and Aurelio Telemann for the S2R+ cell line. We thank Christian Bleischwitz, Eva Boberlin, Jana Kubíková, Katharina Müller, Rebecca Reinig, Gabrielė Ubartaitė, and Xiaohan Zhao for technical assistance provided with some experiments. We thank the Nikon Imaging Center at the University of Heidelberg for access to microscopes. We thank the data storage service SDS@hd supported by the Ministry of Science, Research and the Arts Baden-Württemberg (MWK) and the German Research Foundation (DFG) through the grants INST 35/1314-1 FUGG and INST 35/1503-1 FUGG. We thank Peter Becker, Hüseyin Besir, Julien Béthune, and Doris Höglinger for their insightful comments on the manuscript. This work was funded by the Emmy Noether Program of the German Research Foundation (DFG; JE-827/1-1 to M.J.).

## Author contributions

H.K.S. and J.M. acquired, analyzed, and interpreted the data. M.J. conceived and supervised the project, acquired, analyzed, and interpreted the data, and wrote the manuscript with input from all authors.

## Funding

## Competing interests

The authors declare no competing interests.
