## [Peer Review File · Nature Communications]

REVIEWER COMMENTS

Reviewer #1 (Remarks to the Author):

The paper of Salgania et al. describes a new method called “ReLo” to detect and/or confirm paired protein-protein interactions (PPIs) in live cells. The method is based on co-localization with the bait protein that can be targeted to either the plasma membrane (for cytoplasmic proteins) or reticulum endoplasmic (RE, for nuclear proteins). The method is simple, efficient and specific, with a large panel of applications in many different cell and protein systems.

Still, the work suffers of several major and minor caveats that need to be addressed before considering it for publication.

Major concerns

- Authors claim in the title and the abstract that their method allows to specifically capture direct PPIs. This conclusion is based on the analysis of PPI properties between components of the CCR4-NOT complex, and between subunits of the Arp2/3 complex. More specifically, authors showed that indirect PPIs cannot be revealed in the absence of the intermediary partner. We do not agree with their conclusion: what authors showed is simply the absence of interaction between two partners when a third partner is required. As soon as the third partner is provided (as performed by the authors by co-transfection), authors can reveal the indirect PPI. Therefore, the conclusion is rather the opposite: the method can reveal both direct and indirect PPI, like all other PPI methods. Authors should definitively change their conclusion and present their experiments accordingly.

- The readout of ReLo is an on/off (or qualitative) readout and we agree with the authors that quantifying the amount of colocalized proteins will not obligatorily be of strong utility in this context. Still, one important question is the amount of proteins that is required to obtain a colocalized pattern. No information is provided about the transfection experiments (or very little, see minor concerns below), and one can wonder whether there is a threshold under which positive interactions could not be revealed. Or reversely, a threshold above which false positives could be seen. In any case, authors should test different concentrations, in comparison of endogenous products (for example by using CCR4/NOT or Arp2/3 components) to clearly establish the specificity of their method with regard to the expression level. Along this line, providing inducible expression vectors instead of constitutive high actin-promoter-containing expression vector will be more appropriate and useful.

- There is no statistics. Authors show in supplementary the different patterns in the different transfected cells of one replicate, but the colocalization should be systematically quantified from three different biological replicates (as a percentage of cells showing the colocalization pattern or not).

Minor concerns

- Figure 1: would be nice to have the same panel of Vasa and Oskar constructs systematically tested with either the PH-mCherry or OST4-mCherry system. For example, negative controls like Vasa 463-661 with Oskar 139-240 or 241-387 should be shown with OST4-mCherry.

Although authors claimed that the fusion topology has no incidence (which is surprising), they should also show all these control experiments with alternative fusion topologies in both Vasa and Oskar constructs. Test different expression levels (low/middle/high).

- Figure 2B: bottom panel, Oskar OST4-mCherry does not look like in the ER (in contrast to the upper panel), making the specificity control with Vasa closed less convincing.

- Figure 2D: bottom panel (Aub 4R-to-K), it seems there is some remaining EGFP signal at the plasma membrane. May be in some cases, like this one, it will be useful to quantify the signal.

- Figure 3 and Suppl. Fig. 5: provide the ct-values of QPCR of endogenous products in supplementary. As mentioned previously, test different expression levels, especially in comparison to the endogenous ones.

Since the results show that ReLo reveals indirect interactions as soon as the intermediary partner is provided, merging the result sections corresponding to the Fig.3 (under the same kind of title like "ReLo reveals direct and indirect interactions between different components of a multiprotein complex") will make more sense in the main text.

- Figure 4G: the addition of Cup changes the localization of Bruno-PH-mEGFP (not anymore localized at the plasma membrane), which forbids raising any conclusion for Nanos-mCherry.

- The Methods section is too light. Authors should describe more precisely their transfection experiments (how much plasmid, time of observation after transfection (should be the same for each experiment, and more precise than saying "one or two days"), constructs (do provide the sequence of PH and OST4, what is the linker region if any, etc). For practical reasons, it would also be useful to know whether the colocalization experiments could be imaged on fixed cells.

Reviewer #2 (Remarks to the Author):

New methods for testing PPI are always welcome simply because publications are overwhelmed with false positive results from assays like Y2H or co-IP that could easily mislead following studies for years. Ideally, one wishes to employ an assay to capture specific PPI while avoiding false ones due to method-embedded limitations. While false positives may be detected by alternative methods, lack of interaction due to assay sensitivities could have been ignored. A new method often is examined for criteria like whether the method is more robust than existing methods or overcomes barriers by others. ReLo was designed to have the bait protein relocalized to the plasma membrane or ER in order to redirect the direct interactors to this new site.

ReLo is robust and only requires transfection and standard fluorescence microscopy. It offers qualitative assessment of positive vs. negative outcomes.

Taking one step further, the authors also demonstrated that ReLo was expanded to detect 3-party interaction which is often not addressed in existing methods.

While the advantages of ReLo are well appreciated, a few comments may require the authors to address.

1. The authors may want to emphasize explicitly what ReLo can do that are difficult or impossible for existing PPI methods to deliver. NC probably does not want to see this as another alternative PPI method. For example, it would be useful to inform the readers whether there were proteins tested that could not be relocalized to PM or ER by ReLo.

2. The authors emphasized that ReLo detected conformation-dependent interactions. There is no question that it recapitulated such a phenomenon with the example. However, it had already been known that the mutation could sabotage the interaction and ReLo reached an identical conclusion. Ideally, one would wish to introduce a conformational change following protein expression and test whether such change could abolish the interaction. The Rapamycin and Nutlin experiments were informative but only showed quantitative differences before and after drug treatments. Inexperienced researchers may face a decision-making dilemma when the results became variable from cell to cell.

3. The authors may consider list dis/advantages of existing methods and of course ReLo. It could be extremely helpful for readers to choose ReLo but not others.

Reviewer #3 (Remarks to the Author):

The authors present a novel approach to detect protein-protein interactions (PPI) by which one protein is directed to a membrane by a fused membrane anchoring domain, and a second protein is relocated to the membrane if it interacts with the first protein. Both proteins are fused to different fluorescent tags to follow their location and abundances by fluorescence microscopy. The assays were conducted in *Drosophila* cultured cells but could readily be conducted in other cell types by using appropriate expression vectors and membrane localization signals.

The experiments were well-designed and well-executed and convincingly demonstrate that: most of the known interactions that were tested could be detected; a few known interactions were not detected

indicating that false negatives are possible; both direct and indirect interactions could be detected; interactions dependent on a posttranslational modification (arginine methylation) could be detected; and the assay can be used to reveal non-interacting mutants and molecules that promote or inhibit certain PPI.

The only problems with this paper are several instances of conclusion statements that overreach the data. These are explained below and could be fixed in most cases by re-phrasing the conclusion.

1. From the abstract, "...with ReLo specifically direct but not indirect pairwise PPIs are detected..." Similar statements appear on line 52/53 and line 201 (a subheading). The authors have shown that for two protein complexes expressed in SR2+ cells, direct interactions were detected while indirect interactions were not detected. Use of two complexes is insufficient to prove that ReLo specifically detects only direct PPI. Indeed, the authors also show that a different indirect interaction can be detected when they expressed a bridging protein. There is no reason to believe that other indirect interactions would occasionally be detected through a naturally expressed bridging protein. They may conclude that they did not see indirect interactions involving two specific protein complexes, but they cannot generalize from that observation to conclude that ReLo detects only direct interactions.

2. Intro line 57. "ReLo...enables comprehensive initial description of direct PPI networks." The problem here is the unnecessary and inaccurate word 'comprehensive.' Thus far, no method has proven to be comprehensive in its ability to detect all PPI; most methods have well-characterized false negative rates, not to mention difficulties for scaling up to detect all PPI comprehensively. The paper provides no evidence that ReLo would be any different.

3. "...and we assume that the applicability of Y2H for testing interactions is case/project-dependent and less reliable when testing large proteins from higher eukaryotes." It is well-established that Y2H works for some interactions and not for others. However, the same has been shown for co-AP/MS, PCA, and any number of other assays, and no doubt would apply to ReLo, too. The first half of the assumption implies that this is a unique problem of Y2H and not a problem for ReLo. It is likely that both ReLo and Y2H can detect some interactions that the other cannot. It will require more than a handful compared PPI to determine if one assay's false negative rate is uniquely better than the other. The same is true for "large proteins from higher eukaryotes." It is true that Y2H PPI data is slightly biased against larger ORFs. But it is also true that Y2H has detected interactions with very many large human proteins. It will take more than a handful PPI to determine whether ReLo has any similar bias.

4. This statement, "However, Y2H and PCA are not particularly suitable for the analysis of large and/or unstable proteins because these proteins may be barely expressed or rapidly degraded in a cell" is not supported by the references cited (Lalonde et al. 2008; Cui et al. 2019). There is some evidence from large datasets that Y2H is slightly biased against larger ORFs (e.g., in Rolland 2014). Again, the

demonstration that ReLo works with a small number of large proteins is insufficient to generalize that it works better than PCA or Y2H on larger proteins. This is particularly true of PCA, which also works in cultured cells, involves fusion proteins, and is somewhat more independent of location.

5. The author's solution to the problem appears to be making the two proteins quantifiable with fluorescent tags so that their expression levels or instability can be monitored. This does not increase the chances of detecting interactions with a large or unstable protein, but it does report when the absence of an interaction detection is due to low expression. Similar approaches have been used in other systems, including Y2H – e.g., see Cluet 2020 PMID: 32015065.

Minor comment

Please explain the 'control' in all figures. Many different types of control can be envisioned, though it is not clear which ones are being shown. In Fig 1C, for example, it looks like all cells express a membrane localized mCherry and all cells express a EGFP; so is the 'control' in each case something other than Oskar and Vasa? Is it the vector expressing just MA-mCherry or just EGFP? Please specify in the legends.

Reviewer #4 (Remarks to the Author):

This paper describes the ReLo (relocation) method for detecting and investigating interactions in a cellular context, which can be adapted to identify domains and screen for disrupting mutations. The authors demonstrate that ReLo can detect direct interactions and has other versatilities. ReLo is based on the re-localization of a bait protein to a membrane-anchored protein.

As the authors demonstrate, ReLo has limited utility if the protein of interest has a dominant sorting signal- e.g. nuclear localization signal, which may reduce it broader application. They somewhat overcame the problem with a single nuclear protein by choosing an ER anchoring protein. Will this strategy work with other nuclear proteins? The user will have to determine which anchor works for their protein of interest. They provide multiple examples of protein interactions that demonstrate the procedure works. They validated the system using the Ccr4-Not complex, which has a well characterized domain structure (although I question some of the results, see below). The authors demonstrate it works for most of the examples examined. It will be a new addition to the PPI toolbox. However, I think this work is more appropriate for a specialized journal.

1. The system relies on transfection and likely overexpression of proteins. While the method may have some uses, methods that utilize endogenously expressed proteins would be more reliable.

2. A nice example of how it ReLo can be used to distinguish different conformations was provided. It is unknown ReLo can be applied to other proteins, but nice..

3. The example of Roquin is a bit confusing to me. I don't understand the logic of it interacting with all of the subunits they claim in Ccr4-Not. I wonder if Roq is a sticky protein. Anyway, I won't hold the authors to prior confusing results. However, I have a different reading of the Sgromo 2017 paper. My reading of the paper they cite is that it clearly binds to CAF40. Roq also bound to recombinant pentameric complexes and Not1/2/3 subcomplexes in vitro. I did not see that it bound to 2,3 or Caf1 directly. The results claiming interaction with these proteins (supplement f2) were done by IP of tagged versions of individual subunits of the complex, which would pull down the entire Ccr4-Not complex. I wonder if the authors results with ReLo are in fact picking up indirect interactions or ones bridged by RNA. I looked over the other paper they cite on repressor-Ccr4-Not interactions and there are other instances where the interaction with a single subunit was based by IP from cell extracts. The authors should go back and carefully separate instances where direct, single subunit interactions have been documented versus those inferred by protein subcomplexes or those via co-IP of tagged subunits from cell extracts. This is important to validate their claims.

4. I am surprised at the different localization of Not1 and other subunits of the complex. Is the fusion protein causing mislocalization?

5. superiority to other split protein systems or FRET-based in metazoan cell assays has not been demonstrated

RESPONSE TO THE REVIEWERS

We thank all reviewers for their careful inspection of our manuscript. We hope that, with their help, we were able to improve the manuscript. Our answers are provided in blue color below.

In addition, during the time when the manuscript has been in review, we generated a structural model of the CAF40-Nanos complex using AlphaFold-multimer version 3 and validated the model using ReLo. The CAF40-Nanos interaction is a novel interaction that we identified with ReLo. We included this data as **new Figures 4h and 4i and Supplementary Figure 8** as we think adding this information further strengthens the manuscript. We hope that the reviewers agree with adding these data during the revision.

REVIEWER COMMENTS

Reviewer #1 (Remarks to the Author):

The paper of Salgania et al. describes a new method called "ReLo" to detect and/or confirm paired protein-protein interactions (PPIs) in live cells. The method is based on co-localization with the bait protein that can be targeted to either the plasma membrane (for cytoplasmic proteins) or reticulum endoplasmic (RE, for nuclear proteins). The method is simple, efficient and specific, with a large panel of applications in many different cell and protein systems. Still, the works suffers of several major and minor caveats that need to be addressed before considering it for publication.

Major concerns

- Authors claim in the title and the abstract that their method allows to specifically capture direct PPIs. This conclusion is based on the analysis of PPI properties between components of the CCR4-NOT complex, and between subunits of the Arp2/3 complex. More specifically, authors showed that indirect PPIs cannot be revealed in the absence of the intermediary partner. We do not agree with their conclusion: what authors showed is simply the absence of interaction between two partners when a third partner is required. As soon as the third partner is provided (as performed by the authors by co-transfection), authors can reveal the indirect PPI. Therefore, the conclusion is rather the opposite: the method can reveal both direct and indirect PPI, like all other PPI methods. Authors should definitively change their conclusion and present their experiments accordingly.

In a way, the reviewer is right in saying that with ReLo we detect direct and indirect interactions. Indirect interactions can be detected when we coexpress the bridging factors. This setup could also be considered as two (or more) direct interactions in one experiment.

We used the words "direct" or "indirect" in context of the proteins to be tested in the system. In line with this, we showed that ReLo specifically detects direct interactions. For example, we do not observe the interaction between the indirectly associated subunits NOT3 and CAF1 of the CCR4-NOT complex. The reviewer

explains that this is due to the fact that the third partner is missing. However, the bridging factor NOT1 is not missing, as it is expressed in S2R+ cells. Yet, we do not detect bridging between the ectopic NOT3 and CAF40 and endogenous NOT1. We only observe the bridging if we provide NOT1 also ectopically. All observations led to the conclusion that the cell-endogenous proteins are "invisible" to the system. In fact, this feature of the ReLo assay is not shared by most other PPI methods. For example, in co-IP experiments and proximity labeling methods very often proteins are captured that are not a direct interaction partner of the bait. For many cell-based assays, it has not been tested specifically and thus is unclear.

To avoid confusions, we have now described "indirect interactions" with slightly other words at several instances in the main text. We also added "binary" in the context of "direct" interactions at some instances. We have also slightly rephrased the paragraph describing the bridging experiments. For the specific changes please see the revised manuscript in which the changes were tracked. If the reviewer thinks this is not enough, we will make efforts to improve the description further.

- The readout of ReLo is an on/off (or qualitative) readout and we agree with the authors that quantifying the amount of colocalized proteins will not obligatory be of strong utility in this context. Still, one important question is the amount of proteins that is required to obtain a colocalized pattern. No information is provided about the transfection experiments (or very little, see minor concerns below), and one can wonder whether there is a threshold under which positive interactions could not be revealed. Or reversely, a threshold above which false positives could be seen.

We have now added more details about the transfection protocol. It is essentially according to the instruction manual provided by the manufacturer of the transfection reagent. Please see the revised methods section for details.

We have so far not detected false positive interactions. (This could have been an issue when testing the CCR4-NOT complex or the Arp2/3 complex.) False negatives are conceivable if the expression levels are very low combined with low affinity or if the tagging sterically hinders the binding of the interaction partner. These are possible issues with all cell-based PPIs. Our expression vectors contain linkers between the fluorescent protein and the protein of interest to minimize potential issues.

We cannot say anything about potential thresholds, as we do not know the expression levels of the proteins tested. To determine the in-cell concentration of the proteins is not an easy task (see also our answer below to the minor concern #5 regarding the ct-values). Precisely because we did not know aspects like these, we did rigorous testing of the system with example PPIs (low affinity; multisubunit complexes; etc.) to be able to draw conclusions.

In any case, authors should test different concentrations, in comparison of endogenous products (for example by using CCR4/NOT or Arp2/3 components) to clearly establish the specificity of their method with regard to the expression level.

To address the reviewer's questions, we have now tested several variations to the transfection protocol. Please see **Rebuttal Figure 1** below. Reducing the DNA amount (and accordingly the transfection reagent) during transfection (from standard 600 ng to 300, 150, to 75 ng total DNA) did not change the average expression

levels observed per cell. However, using only 150 or 75 ng of DNA resulted in a reduced transfection efficiency (number of cells transfected). Lowering the DNA amount while keeping the transfection reagent at standard amounts also did not change the expression levels in the cells. Here, the transfection efficiency declined when using the lowest DNA amount (75 ng). Taken together, the transfection protocol is highly robust and changing it did not change protein expression levels of a given protein combination. Consequently, we detected the interaction tested in all experiments to a comparable level.

We performed transient transfections, which results in various expression levels across cells and imaging of cells with different fluorescence intensities. During the course of our experiments, we did not observe qualitative differences in the detection of interactions. The observation of interactions seems relatively independent of the signal intensities. However, the relative expression of the mCherry vs. the EGFP construct within one cell matters. If for example the membrane-anchored construct is much lower expressed as compared to the non-anchored construct, efficient relocalization of the non-anchored protein might not be detectable. This is a potential issue common to all PPI methods. But looking at many cells (with slightly different expression levels and ratios) would still allow to draw a conclusion.

To determine the concentration of endogenous proteins and to compare these levels to the ectopically expressed construct is highly challenging. We cannot use Western blot to address this question as we perform transient transfection, upon which only a subset of the cells expresses the proteins of interest (see also below).

Along this line, providing inducible expression vectors instead of constitutive high actin-promoter-containing expression vector will be more appropriate and useful.

From our experiments we conclude that ReLo works because we use a strong promoter (especially with respect to the detection of direct interactions). We did not experience issues with this expression system. As such, we would need more information from the reviewer in order to comprehend why an inducible expression vector is more appropriate and useful. A constitutive expression system is simpler and faster as compared to an inducible one, as it does not require an extra reagent and eventually extra incubation time for the induction of expression. We think that the strength of the ReLo assay is the speed and simplicity.

- There is no statistics. Authors show in supplementary the different patterns in the different transfected cells of one replicate, but the colocalization should be systematically quantified from three different biological replicates (as a percentage of cells showing the colocalization pattern or not).

Whether or not a protein relocalized with another was typically clearly visible. With the data presented in Supplementary Figures 2 and 3 we aimed to give an impression of what is observed at the microscope. We now repeated these experiments two more times independently (even by two different people) (see **Rebuttal Figure 2**) and provide statistical information in the legends to Supplementary Figures 2 and 3.

Minor concerns

- Figure 1: would be nice to have the same panel of Vasa and Oskar constructs

systematically tested with either the PH-mCherry or OST4-mCherry system. For example, negative controls like Vasa 463-661 with Oskar 139-240 or 241-387 should be shown with OST4-mCherry.

We have now added the experiments the reviewer asked as **new Supplementary Figure 1e**. We have also added the Vasa mutant experiment to the main Figure 1e. All experiments are consistent and independent of which membrane anchor was used.

Although authors claimed that the fusion topology has no incidence (which is surprising), they should also show all these control experiments with alternative fusion topologies in both Vasa and Oskar constructs. Test different expression levels (low/middle/high).

We believe there is a misunderstanding here. In our manuscript, we did not claim that the fusion topology has no influence. Instead, we aimed to explain that the PH domain acts as a plasma membrane anchoring domain independent of its position within the fusion construct (N- or the C-terminus). This is in contrast to the OST4 anchor, for example, which can only be used as an N-terminal fusion for its localization to the ER. Of course, fusions might have an impact on the accessibility of protein regions for binding to their partners - possibly also in the ReLo assay. To reduce the risk of steric hindrance, all constructs contain a linker sequence between the fluorescence tag and the proteins of interest.

- Figure 2B: bottom panel, Oskar OST4-mCherry does not look like in the ER (in contrast to the upper panel), making the specificity control with Vasa closed less convincing.

The localization of OST4-Oskar differs slightly depending on the expression levels. In cells with strong signal intensity, Oskar forms granular structures around the nucleus. In cells with weaker signal, Oskar localization looks more similar to the OST4-mCherry control ER localization. We added an Oskar control experiment to Figure 2b. Please also compare with Figure 1e and Supp Fig 1e.

What we think matters and what we look at is whether the prey protein (Vasa) relocated to the position of the localized bait (Oskar). Vasa closed is equally distributed throughout the cytoplasm of the cell and not enriched at the position of Oskar. This is in contrast to Vasa open, which fully "merges" with Oskar. Our conclusion is based on this strong difference.

- Figure 2D: bottom panel (Aub 4R-to-K), it seems there is some remaining EGFP signal at the plasma membrane. May be in some cases, like this one, it will be useful to quantify the signal.

We have now added line plots to Figure 2d to better visualize the signal distribution across the cells.

- Figure 3 and Suppl. Fig. 5: provide the ct-values of QPCR of endogenous products in supplementary. As mentioned previously, test different expression levels, especially in comparison to the endogenous ones.

Determining the ct-values by qPCR allows to conclude about mRNA levels. This method will not provide information about the protein levels. Furthermore, neither qPCR nor Western blot (to detect protein levels) analysis of molecules from cell lysates will be helpful, as the cell population is a mix of transfected and non-transfected cells. To compare the ectopic expression levels to the endogenous ones would require experiments that detect the protein levels in each individual cell. We think that such elaborate experiment is beyond the scope of the current study. We would like to emphasize that we have shown how the ReLo assay can be used to address various questions regarding the characterization of PPIs and that as such it is a simple and quick screening method.

Since the results show that ReLo reveals indirect interactions as soon as the intermediary partner is provided, merging the result sections corresponding to the Fig.3 (under the same kind of title like “ReLo reveals direct and indirect interactions between different components of a multiprotein complex”) will make more sense in the main text.

We tried to clarify a bit more on direct vs. indirect interactions (see above). We want to emphasize on the description of the complex topology rather than the indirect interactions per se and therefore would like to keep it as a separate section.

- Figure 4G: the addition of Cup changes the localization of Bruno-PH-mEGFP (not anymore localized at the plasma membrane), which forbids raising any conclusion for Nanos-mCherry.

Bruno is a protein that tends to form granules in the cells - also when using a fusion to a PH domain. In the original image, Bruno showed a distinct dotted localization (and some membrane localization) and Nanos did not colocalize specifically with Bruno, which allowed us to conclude that Bruno and Nanos did not interact in the presence of Cup. For better clarity, we have now replaced the image in question with another representative one, in which most of the Bruno protein localized to the membrane. Also in this cell, Nanos did not colocalize with Bruno.

- The Methods section is too light. Authors should describe more precisely their transfection experiments (how much plasmid, time of observation after transfection (should be the same for each experiment, and more precise than saying “one or two days”), constructs (do provide the sequence of PH and OST4, what is the linker region if any, etc).

We have now added more detailed information to the Methods sections. We have also provided now the sequence of the PH domain, the OST4 miniprotein and the linkers within the Supplementary Information.

For practical reasons, it would also be useful to know whether the colocalization experiments could be imaged on fixed cells.

We have tested one fixation protocol for cultured cells with the CAF1-CCR4 interaction: 48 h after the transfection, the medium was removed and cells were washed once with 1xPBS. The cells were then fixed for 15 mins at room temperature with 500 µl of 4% paraformaldehyde (in 1xPBS) followed by three washing steps with

500 μ l of 1xPBS each. The cells were left in 500 μ l of 1xPBS and subsequently imaged.

However, we observed issues with this protocol (**Rebuttal figure 3**). In general, the signal intensity was very poor as compared to live imaging and it was hard to image cells at all. Furthermore, while we detected the relocalization event in the experiment, we hardly detected the membrane localization in the controls. It seems that the fixation method we used disrupted the membrane integrity. Nevertheless, we assume that fixation might be possible but it would require optimization of the fixation protocol. (We have only experience with fixing *Drosophila* ovaries or embryos.)

Reviewer #2 (Remarks to the Author):

New methods for testing PPI are always welcome simply because publications are overwhelmed with false positive results from assays like Y2H or co-IP that could easily mislead following studies for years. Ideally, one wishes to employ an assay to capture specific PPI while avoiding false ones due to method-embedded limitations. While false positives may be detected by alternative methods, lack of interaction due to assay sensitivities could have been ignored. A new method often is examined for criteria like whether the method is more robust than existing methods or overcomes barriers by others. ReLo was designed to have the bait protein relocalized to the plasma membrane or ER in order to redirect the direct interactors to this new site.

ReLo is robust and only requires transfection and standard fluorescence microscopy. It offers qualitative assessment of positive vs. negative outcomes.

Taking one step further, the authors also demonstrated that ReLo was expanded to detect 3-party interaction which is often not addressed in existing methods.

While the advantages of ReLo are well appreciated, a few comments may require the authors to address.

1. The authors may want to emphasize explicitly what ReLo can do that are difficult or impossible for existing PPI methods to deliver. NC probably does not want to see this as another alternative PPI method.

For a better overview, we have now prepared a **new Supplementary Table 1** comparing various cell-based PPI methods, including ReLo, Y2H, PCA, and FRET-based assays.

For example, it would be useful to inform the readers whether there were proteins tested that could not be relocalized to PM or ER by ReLo.

As described in the manuscript, nuclear proteins could not be localized efficiently to the plasma membrane using the PH domain fusion (See Supplementary Figure 1c). A fusion to the OST4 miniprotein is known for its capacity to retain nuclear proteins within the cytoplasm and this has already been applied in yeast cells in Möckli et al., 2007; BioTechniques. Consistently, all nuclear proteins we have tested so far were anchored to the ER using the fusion to the OST4 miniprotein. In addition to Short Oskar, we have tested a few other nuclear proteins and all were successfully anchored to the ER (examples shown in **Rebuttal Figure 4**). Taken together, we did not observe any issues other than the one already described in the manuscript.

2. The authors emphasized that ReLo detected conformation-dependent interactions. There is no question that it recapitulated such a phenomenon with the example. However, it had already been known that the mutation could sabotage the interaction and ReLo reached an identical conclusion. Ideally, one would wish to introduce a conformational change following protein expression and test whether such change could abolish the interaction.

We are not sure if we understand the reviewer's concern/suggestion. In our manuscript, we presented experiments with two different "conformational" mutations:

The Vasa K295N (Vasa-open) mutation is unable to bind to ATP and retains the protein in an open conformation. This conformation is able to bind to Short Oskar. The second, Vasa E400Q (Vasa-closed) mutation interferes with the product release upon ATP hydrolysis and retains the protein in a closed conformation. This closed conformation is unable to interact with Short Oskar. None of these mutations lie in the interface of the Oskar-Vasa complex.

If this does not answer the question, we would be happy if the reviewer rephrased the question and we will make efforts to answer the question appropriately.

Independent of that, we realized that the numbering of the Vasa-open mutation was incorrect (K282N). We now have corrected it at all instances to K295N.

The Rapamycin and Nutlin experiments were informative but only showed quantitative differences before and after drug treatments. Inexperienced researchers may face a decision-making dilemma when the results became variable from cell to cell.

With the experiments presented we aimed to show that the assay is responsive to drug treatment per se. To perform drug screening experiments, the additional establishment of an appropriate setup might be required. I.e., expression of the two desired interaction partners from one plasmid, applying automated imaging and analyses, etc. We have discussed these possibilities in the discussion section.

3. The authors may consider list dis/advantages of existing methods and of course ReLo. It could be extremely helpful for readers to choose ReLo but not others.

We have now prepared a **new Supplementary Table 1** comparing various cell-based PPI methods (see above).

Reviewer #3 (Remarks to the Author):

The authors present a novel approach to detect protein-protein interactions (PPI) by which one protein is directed to a membrane by a fused membrane anchoring domain, and a second protein is relocated to the membrane if it interacts with the first protein. Both proteins are fused to different fluorescent tags to follow their location and abundances by fluorescence microscopy. The assays were conducted in *Drosophila* cultured cells but could readily be conducted in other cell types by using appropriate expression vectors and membrane localization signals.

The experiments were well-designed and well-executed and convincingly demonstrate that: most of the known interactions that were tested could be detected; a few known interactions were not detected indicating that false negatives are possible; both direct and indirect interactions could be detected; interactions dependent on a posttranslational modification (arginine methylation) could be detected; and the assay can be used to reveal non-interacting mutants and molecules that promote or inhibit certain PPI.

The only problems with this paper are several instances of conclusion statements that overreach the data. These are explained below and could be fixed in most cases by re-phrasing the conclusion.

1. From the abstract, "...with ReLo specifically direct but not indirect pairwise PPIs are detected...." Similar statements appear on line 52/53 and line 201 (a subheading). The authors have shown that for two protein complexes expressed in SR2+ cells, direct interactions were detected while indirect interactions were not detected. Use of two complexes is insufficient to prove that ReLo specifically detects only direct PPI. Indeed, the authors also show that a different indirect interaction can be detected when they expressed a bridging protein. There is no reason to believe that other indirect interactions would occasionally be detected through a naturally expressed bridging protein. They may conclude that they did not see indirect interactions involving two specific protein complexes, but they cannot generalize from that observation to conclude that ReLo detects only direct interactions.

In an ongoing study in the lab, we have used the ReLo assay to screen pairwise interactions between 25 proteins involved in a common pathway. We further validated all detected interactions with other methods, and in no instance, we detected an "indirect" interaction. The same is true for a couple of ongoing collaboration projects. These findings made us really confident that the assay is very robust and reliable. Having these data in mind, we probably were overexcited in our phrasing and generalization.

As suggested by the reviewer, we rephrased some statements throughout the manuscript. For example, in the last two paragraphs of the results section entitled "ReLo reveals direct binary interactions", we discuss now "Nevertheless, we cannot exclude that occasionally, indirect interactions are detected through a naturally expressed bridging protein." and "As both Y2H and ReLo worked well to map the Oskar-Vasa interaction, together the data suggest that the applicability of Y2H for testing interactions is case/project dependent. ReLo could be expected to be more reliable than Y2H for testing large proteins from higher eukaryotes, although a larger scale study would be required to fully test this." For additional changes please see

the revised version of the manuscript in which the changes were tracked.

2. Intro line 57. “ReLo...enables comprehensive initial description of direct PPI networks.” The problem here is the unnecessary and inaccurate word ‘comprehensive.’ Thus far, no method has proven to be comprehensive in its ability to detect all PPI; most methods have well-characterized false negative rates, not to mention difficulties for scaling up to detect all PPI comprehensively. The paper provides no evidence that ReLo would be any different.

In the phrase “ReLo...enables comprehensive initial description of direct PPI networks.” we intended to use the word "comprehensive" with respect to the various different conclusions one can obtain by studying one particular interaction (of a given PPI network) using one single assay. We did not intend to use the word to state that all PPIs within a network can be found. We have now replaced the word "comprehensive" with "thorough" and hope this is appropriate.

3. “...and we assume that the applicability of Y2H for testing interactions is case/project-dependent and less reliable when testing large proteins from higher eukaryotes.” It is well-established that Y2H works for some interactions and not for others. However, the same has been shown for co-AP/MS, PCA, and any number of other assays, and no doubt would apply to ReLo, too. The first half of the assumption implies that this is a unique problem of Y2H and not a problem for ReLo. It is likely that both ReLo and Y2H can detect some interactions that the other cannot. It will require more than a handful compared PPI to determine if one assay’s false negative rate is uniquely better than the other. The same is true for “large proteins from higher eukaryotes.” It is true that Y2H PPI data is slightly biased against larger ORFs. But it is also true that Y2H has detected interactions with very many large human proteins. It will take more than a handful PPI to determine whether ReLo has any similar bias.

We aimed to discuss the differential findings obtained by comparing Y2H with the ReLo assay for two given examples (Oskar-Vasa and CCR4-NOT complex subunits). To be clearer, we have now specified the discussion of this data to " As both Y2H and ReLo worked well to map the Oskar-Vasa interaction, together the data suggest that the applicability of Y2H for testing interactions is case/project-dependent and probably less reliable than ReLo when testing large proteins from higher eukaryotes."

As mentioned in an answer to reviewer #2, we have now prepared a **new Supplementary Table 1** comparing various cell-based PPI methods, including ReLo, Y2H, PCA, and FRET-based assays.

4. This statement, “However, Y2H and PCA are not particularly suitable for the analysis of large and/or unstable proteins because these proteins may be barely expressed or rapidly degraded in a cell” is not supported by the references cited (Lalonde et al. 2008; Cui et al. 2019).

The full original sentence is: "However, Y2H and PCA are not particularly suitable for the analysis of large and/or unstable proteins because these proteins may be barely expressed or rapidly degraded in a cell resulting in unreliable, mostly false-negative results." With this sentence we tried to make the point that unstable proteins are an

issue as they result in false-negative data. Protein instability leading to false-negative data are discussed in the papers cited.

For better clarity, we have now rephrased the sentence to: "However, Y2H and PCA are not particularly suitable for the analysis of potentially unstable proteins, as they are barely expressed or rapidly degraded in a cell, resulting in unreliable, mostly false-negative results."

There is some evidence from large datasets that Y2H is slightly biased against larger ORFs (e.g., in Rolland 2014). Again, the demonstration that ReLo works with a small number of large proteins is insufficient to generalize that it works better than PCA or Y2H on larger proteins. This is particularly true of PCA, which also works in cultured cells, involves fusion proteins, and is somewhat more independent of location.

As mentioned above, in an ongoing study in the lab, we have used the ReLo assay to screen pairwise interactions between 25 proteins of a common pathway. 11 of these proteins are larger than 90 kDa and we have had severe issues using Y2H as initial screening method to define a direct PPI network. This is why we believe large proteins with substantial amount of predicted structural disorder causes severe issues in yeast but not in higher eukaryotic cells. We agree with the reviewer that this particular issue should not apply to PCA. However, PCA is less straightforward as compared to the ReLo assay, as it requires a proper orientation or steric accessibility for the split protein to be reconstituted. In this sense, ReLo is less dependent on where the fluorescent protein (half) is located within the fusion protein. Furthermore, if a PPI is not detected with PCA, it remains an open question if the protein is expressed at all. This could be a strong issue when working with structurally complex proteins. And finally, PCA would probably not allow to perform bridging experiments to describe the architecture of protein complexes.

5. The author's solution to the problem appears to be making the two proteins quantifiable with fluorescent tags so that their expression levels or instability can be monitored. This does not increase the chances of detecting interactions with a large or unstable protein, but it does report when the absence of an interaction detection is due to low expression. Similar approaches have been used in other systems, including Y2H – e.g., see Cluet 2020 PMID: 32015065.

We thank the reviewer for pointing at the Cluet 2020 paper! It has missed our attention. While indeed the Y2H method described in the paper allows to monitor protein expression levels simultaneously with the reporter expression, it is unclear if the assay can be used for nuclear and membrane proteins. Also, it is unclear if bridging experiments can be performed to describe the topology of a complex. In conclusion, we think that ReLo combines several advantages.

We now referred to the Cluet paper (and several others) in the new Supplementary Table 1.

Minor comment

Please explain the 'control' in all figures. Many different types of control can be envisioned, though it is not clear which ones are being shown. In Fig 1C, for example, it looks like all cells express a membrane localized mCherry and all cells express a EGFP; so is the 'control' in each case something other than Oskar and

Vasa? Is it the vector expressing just MA-mCherry or just EGFP? Please specify in the legends.

The "Control" always indicated coexpression of a specific construct with the same vector as used for an experiment but lacking an insert. We have now described the "control" in all figure legends.

Reviewer #4 (Remarks to the Author):

This paper describes the ReLo (relocation) method for detecting and investigating interactions in a cellular context, which can be adapted to identify domains and screen for disrupting mutations. The authors demonstrate that ReLo can detect direct interactions and has other versatilities. ReLo is based on the re-localization of a bait protein to a membrane-anchored protein.

As the authors demonstrate, ReLo has limited utility if the protein of interest has a dominant sorting signal- e.g. nuclear localization signal, which may reduce its broader application. They somewhat overcame the problem with a single nuclear protein by choosing an ER anchoring protein. Will this strategy work with other nuclear proteins? The user will have to determine which anchor works for their protein of interest.

Please also see above for an answer to question 1 of reviewer #2. A fusion to the OST4 miniprotein is known for its capacity to retain nuclear proteins within the cytoplasm and this has already been described for yeast cells in Möckli et al., 2007; BioTechniques. As mentioned above, in addition to Short Oskar, we have tested a few other nuclear proteins and all were successfully anchored to the ER (examples shown in **Rebuttal Figure 4**). Although only a few have been tested, we think the selection is representative.

We have now slightly modified the relevant sentence introducing the OST4 anchor (page 6): "Hence, we selected the minimembrane protein subunit 4 of the yeast oligosaccharyltransferase complex (OST4), which was previously used as N-terminal fusion protein to localize a nuclear protein to the endoplasmic reticulum (ER) 26,27 (Supplementary Fig. 1d)."

They provide multiple examples of protein interactions that demonstrate the procedure works. They validated the system using the Ccr4-Not complex, which has a well characterized domain structure (although I question some of the results, see below). The authors demonstrate it works for most of the examples examined. It will be a new addition to the PPI toolbox. However, I think this work is more appropriate for a specialized journal.

1. The system relies on transfection and likely overexpression of proteins. While the method may have some uses, methods that utilize endogenously expressed proteins would be more reliable.

We do not think that endogenously expressed proteins will result in more reliable data. We think that ReLo works because of an overexpression system. We think that the strength of the ReLo assay lies in the observation that PPIs that are observed are direct (pairwise) interactions. In our experience, findings made with ReLo are strongly encouraging to perform subsequent biochemical and genetic experiments.

2. A nice example of how it ReLo can be used to distinguish different conformations was provided. It is unknown ReLo can be applied to other proteins, but nice..

We assume that as long as mutations that "lock" a protein in a specific conformation are well characterized, the assay can also be used to study conformation-specific

interactions to proteins other than DEAD-box RNA helicases.

3. The example of Roquin is a bit confusing to me. I don't understand the logic of it interacting with all of the subunits they claim in Ccr4-Not. I wonder if Roq is a sticky protein. Anyway, I won't hold the authors to prior confusing results.

We found that Roquin interacts with the subunits CAF40, CCR4, NOT1, and NOT3 but not with CAF1 or NOT2. Roquin did not bind all subunits in our assay. We think the PPIs we found with Roquin are specific for the following reasons: We also tested the interactions of the subunits of the CCR4-NOT complex to other RNA binding proteins that are very likely "sticky", such as Smaug and the almost fully disordered protein Cup (Pekovic et al., 2023; Supplemental Figure 7E). However, we did not detect Cup binding to any of the CCR4-NOT complex subunits and we identified only the NOT3 subunit to bind to Smaug.

It is a bit difficult to understand the term "sticky" within the cellular context. We only know its use in co-purification methods, where a "sticky" protein tends to bind unspecifically to the beads or in the negative control experiment.

However, I have a different reading of the Sgromo 2017 paper. My reading of the paper they cite is that it clearly binds to CAF40. Roq also bound to recombinant pentameric complexes and Not1/2/3 subcomplexes in vitro. I did not see that it bound to 2,3 or Caf1 directly. The results claiming interaction with these proteins (supplement f2) were done by IP of tagged versions of individual subunits of the complex, which would pull down the entire Ccr4-Not complex. I wonder if the authors results with ReLo are in fact picking up indirect interactions or ones bridged by RNA.

We have now specified the previously reported data on Roquin in the main text. The reviewer is right that a direct interaction between Roquin and CAF1 has not been shown. In conclusion, our data are consistent with previous data. In addition, we identified CCR4 as novel Roquin binding partner.

We have no evidence that interactions in the ReLo assay are bridged by RNA. We have tested many RNA binding proteins (Bam, Nanos, Smaug, Cup, MARF1, Bruno) and we find specific results for each of them.

I looked over the other paper they cite on repressor-Ccr4-Not interactions and there are other instances where the interaction with a single subunit was based by IP from cell extracts. The authors should go back and carefully separate instances where direct, single subunit interactions have been documented versus those inferred by protein subcomplexes or those via co-IP of tagged subunits from cell extracts. This is important to validate their claims.

We have now further specified the previous data on the interaction between Nanos and the CCR4-NOT complex in the main text (lines 283 ff, pages 13 and 14). The conclusions remain unchanged.

4. I am surprised at the different localization of Not1 and other subunits of the complex. Is the fusion protein causing mislocalization?

In the control data set shown in Supplementary Fig. 4B we show that NOT1 and NOT3 fused to mEGFP localize exclusively in the cytoplasm, while all other subunits

localize to both cytoplasm and nucleus in S2R+ cells. The localization of the individual subunits is fully consistent throughout all our experiments. In all cases the mEGFP fusion is N-terminal to the subunit.

5. superiority to other split protein systems or FRET-based in metazoan cell assays has not been demonstrated

As mentioned above, we have now prepared a **new Supplementary Table 1** comparing various cell-based PPI methods, including ReLo, Y2H, PCA, and FRET-based assays. In conclusion, we think that the superiority of the ReLo assay over PCA or FRET assays lies in its speediness and simplicity and its applicability to define the topology of complexes. We think that all these are important aspects when deciding on which cell-based PPI test to choose for a study.

FIGURES TO THE REVIEWERS

Rebuttal Figure 1. Variation of the both DNA amount and transfection reagent or DNA amount only during the transfection.

The expression levels remain unchanged.

All 31 cells show relocation

Rebuttal Figure 2a. Second replicate related to Supplementary Figure 2.

23 cells out of 24 show relocation

Rebuttal Figure 2b. Third replicate related to Supplementary Figure 2.

All 45 cells show relocalization

Rebuttal Figure 2c. Second replicate related to Supplementary Figure 3a.

All 20 cells show relocation

Rebuttal Figure 2d. Third replicate related to Supplementary Figure 3a.

All 30 cells show relocation

Rebuttal Figure 2e. Second replicate related to Supplementary Figure 3b.

All 23 cells show relocation

Rebuttal Figure 2f. Third replicate related to Supplementary Figure 3b.

Rebuttal Figure 3. Testing fixation of the cells.

"[Redacted]"

Reviewers' comments:

Reviewer #1 (Remarks to the Author):

I am surprised that the different concentrations of DNA did not affect the level of expression but only the proportion of transfected cells. With this regard, what is the percentage of cells co-transfected on average? This information could be mentioned in the M&M section. I still believe that westerns could be performed by selecting fluorescent cells, and I would like to have these data before accepting the manuscript. I consider it is important to know the level of expression of fusion proteins when compared to the endogenous ones.

Reviewer #2 (Remarks to the Author):

In response to comments raised previously, the authors have added a table that compared ReLo and existing popular PPI assays in terms of their advantages and disadvantages. It was surprising, however, that methods mentioned in the text (lines 30-36) were not included in the table.

Previously, point 2 questioned the statement of the detection of conformational mutations. Such a statement may be taken as that the ReLo method could detect conformational differences of a protein. The tested mutation was already known to abolish the interaction prior to the ReLo experiment. In other words, other methods would have done a similar job. It would have been rather surprising if ReLo reached a conclusion different from the reported conclusion. Therefore, the result was not reserved for ReLo.

The authors assured that all tested proteins were re-localized to targeted sites. Indeed, the nuclear proteins of Piwi and ZFP280 became associated with the nuclear envelope and ER upon fusion with OST4. One would think that some proteins might have strong intrinsic localization signals/motifs so that such fusions could result in a serious competition in subcellular localization. The authors might want to consider, e.g., in Discussion, that removal of intrinsic localization signal could help enhance the re-localization as intended.

Reviewer #3 (Remarks to the Author):

Most of the reviewers concerns have been addressed adequately and this paper now appears ready for publication in Nature Communication.

However, I would encourage the authors to consider the following comments.

The authors are still stating far-reaching conclusions about other assays that are not supported by actual data. And these conclusions are not important for the paper, so it is not clear why they are being stated in this way.

One problem is that the authors are using “Y2H” (yeast two-hybrid) to refer to both the split ubiquitin system and the original transcription based yeast two-hybrid system. Most readers (and authors) consider Y2H to refer to only the latter. They are different assays, each with their own advantages and disadvantages, and volumes of data.

The authors used the split ubiquitin system to directly compare with ReLo for two complexes. Thus, they can legitimately say that the “...ReLo assay appears to be successful regardless of the length of the protein of interest, which, in our experience, is a big advantage over the yeast split ubiquitin systems”, but they cannot say that about the Y2H system. The entire paragraph starting on line 246 should use ‘yeast split ubiquitin assay’ rather than Y2H. In the case of the Arp2/3 complex they cited another paper that actually did use Y2H, so it is appropriate to call it Y2H there.

As mentioned previously, the citations 7 and 10 do not contain data that supports this statement: “However, standard Y2H and PCA assays are not particularly suitable for the analysis of potentially unstable proteins, as they are barely expressed or rapidly degraded in a cell, resulting in unreliable, mostly false-negative results.” Those papers are reviews, and even they do not cite data to support that statement. The authors may make such a statement as a speculation or hypothesis, but if they intend to state it as fact, they must cite primary data to support it.

Finally, Supplementary Table 1 in its current form is a little cheesy. First, it introduces a new use for the terms ‘direct’ and ‘indirect’, which are different from their definitions throughout the paper. Second, it combines Y2H and split ubiquitin into the same column and they do not share the same comparative features. Third, in Experimental Setup, is it true that the yeast assays require an incubator and the mammalian cells do not? Or the yeast assay requires a camera and the microscope assays do not? Neither absolutely ‘require’ a camera as you could just look at the results and record them. These are not noteworthy differences. Fourth, “Possible source for false negative/positive results” appears to allow for speculation, so here is some for ReLo: one or the other protein may locate to the wrong

compartment; ectopic overexpression may allow for interaction between proteins that would not normally interact (same for several of the assays listed).

Reviewer #4 (Remarks to the Author):

The authors made a good faith effort to address most of my concerns. I am still not fully convinced it is appropriate for the journal, but the revised version is more scientifically sound.

Point-by point response to the Reviewers' comments

We are grateful to the reviewers for their additional input and we have now further improved the manuscript with their help. First, we performed experiments to determine the expression levels of endogenous and overexpressed subunits of the CCR4-NOT complex. Second, we have thoroughly revised the Supplementary Table 1 and hope that it is now in a form that can serve as a useful overview of cell-based PPI methods. Finally, we have revised some statements in the main text and added more information in the Methods part. We hope that with this further improvement, the reviewers will agree that the manuscript is suitable for publication in Nature Communications.

Reviewer #1 (Remarks to the Author):

I am surprised that the different concentrations of DNA did not affect the level of expression but only the proportion of transfected cells.

We assume that apparently only the transfection efficiency is altered and not the expression levels per cell because we select the cells by their fluorescence signal. For example, if we use low amounts of DNA, there may be cells with low expression that we consider as not transfected due to signals at the detection limit. Alternatively, we could imagine that we achieve already very high expression levels with one copy of plasmid by using the Actin 5 promotor, and perhaps more than one copy of the plasmid does not further enhance the expression. The latter scenario might be explained by toxicity of too high expression levels, resulting in apoptotic cells, which we do not consider in our image analysis.

With this regard, what is the percentage of cells co-transfected on average? This information could be mentioned in the M&M section.

The average co-transfection efficiency is approx. 30%. We now include this information in the Methods section.

I still believe that westerns could be performed by selecting fluorescent cells, and I would like to have these data before accepting the manuscript. I consider it is important to know the level of expression of fusion proteins when compared to the endogenous ones.

To address the question of how the ectopic levels compare to the endogenous expression levels of a given protein, we decided to use a western blot approach and planned to analyse all subunits of the CCR4-NOT complex, for which we also determined the pairwise interaction network (Figure 3). We obtained sera and monoclonal antibodies from the lab of Elmar Wahle, the only lab that raised antibodies against all subunits of the *Drosophila* complex. Due to the retirement of Elmar Wahle, the lab is now closed. Unfortunately, during the course of our experiments, we encountered difficulties with some of the sera.

1. With the sera against CCR4 and CAF1, we could not detect a specific signal. The Wahle lab used to affinity-purify these sera using purified CCR4 and CAF1 proteins, but they did not keep any affinity-purified antibodies due to the closure of the lab. We obtained all of the remaining purified CCR4-CAF1 heterodimer from the Wahle lab and attempted to perform the affinity purification of the anti-CCR4 and anti-CAF1 sera in

our lab. This purification was not successful, probably because the amount of the protein complex used to purify the sera was not high enough.

2. The anti-CAF40 serum has lost its activity, an observation that was also made by the last members of the Wahle lab.

3. The monoclonal anti-NOT1 antibody lost its activity during the course of the experiments and we obtained only one replicate for the analysis, which revealed an approximately 8-fold overexpression.

Because of the issues, we could only use the sera against NOT2, and NOT3 to perform the western blot analysis in several replicates (3-4 x). We loaded a 2-fold serial dilution of the samples and determined the relative expression levels by quantifying the signals using Fiji. Subsequently, the obtained expression ratios were corrected for the transfection efficiencies, which were determined in parallel by microscopy. With these two example proteins, the overexpression varied between 2- and 27-fold. We included these data in the manuscript as **Supplementary Figure 5**.

Although we can only offer these two examples, we hope that they satisfy what the reviewer wanted to see.

Reviewer #2 (Remarks to the Author):

In response to comments raised previously, the authors have added a table that compared ReLo and existing popular PPI assays in terms of their advantages and disadvantages. It was surprising, however, that methods mentioned in the text (lines 30-36) were not included in the table.

For the comparison table, we have chosen only the most commonly used cell-based assays (Y2H, complementation assays, FRET-based assays) because these are the assays that readers may have heard of when looking for a cell-based PPI assay. As we have already discussed the advantages and disadvantages of other existing translocation methods in the Introduction section, we have not included them in the Supplementary Table 1. The intention was to focus on clarity and not to hide information.

Previously, point 2 questioned the statement of the detection of conformational mutations. Such a statement may be taken as that the ReLo method could detect conformational differences of a protein. The tested mutation was already known to abolish the interaction prior to the ReLo experiment. In other words, other methods would have done a similar job. It would have been rather surprising if ReLo reached a conclusion different from the reported conclusion. Therefore, the result was not reserved for ReLo.

We agree that conformation-dependent interactions can also be analyzed with other methods. We showed data using ReLo assays, because we thought that some readers might find it interesting to know. In fact, Reviewer 4 pointed out that "a nice example of how it ReLo can be used to distinguish different conformations was provided." (see their initial comments).

The mutations that we introduced into Vasa affect ATP binding and ADP*P release, respectively, and are not on the surface that contacts Oskar. To clarify this point, we have added a sentence in the main text: "Neither mutation is located in the Vasa-Oskar binding interface."

The authors assured that all tested proteins were re-localized to targeted sites. Indeed, the nuclear proteins of Piwi and ZFP280 became associated with the nuclear envelope and ER upon fusion with OST4. One would think that some proteins might have strong intrinsic localization signals/motifs so that such fusions could result in a serious competition in subcellular localization. The authors might want to consider, e.g., in Discussion, that removal of intrinsic localization signal could help enhance the re-localization as intended.

We thank the reviewer for this suggestion and have added the following sentence to the Discussion: "For proteins with strong intrinsic localization signals, it may be necessary to remove such signals before testing for an interaction-induced relocalization."

Reviewer #3 (Remarks to the Author):

Most of the reviewers concerns have been addressed adequately and this paper now appears ready for publication in Nature Communication.

However, I would encourage the authors to consider the following comments.

The authors are still stating far-reaching conclusions about other assays that are not supported by actual data. And these conclusions are not important for the paper, so it is not clear why they are being stated in this way.

One problem is that the authors are using "Y2H" (yeast two-hybrid) to refer to both the split ubiquitin system and the original transcription based yeast two-hybrid system. Most readers (and authors) consider Y2H to refer to only the latter. They are different assays, each with their own advantages and disadvantages, and volumes of data.

Regarding the "classical" Y2H assay and the split-ubiquitin Y2H assay, we are aware that the underlying interaction principles are different between the two assays. We placed them in a similar "category" because in both cases the proteins are expressed in yeast, and the readout is indirect (expression activation of one or more reporter genes). The yeast system is likely to cause expression/stability problems when working with structurally complex proteins from higher eukaryotes. We thought this was the most relevant point for the comparison with ReLo. We now differentiate the statements with respect to a more specific description of which Y2H assay is discussed in each text section.

The authors used the split ubiquitin system to directly compare with ReLo for two complexes. Thus, they can legitimately say that the "...ReLo assay appears to be successful regardless of the length of the protein of interest, which, in our experience, is a big advantage over the yeast split ubiquitin systems", but they cannot say that about the Y2H system.

The entire paragraph starting on line 246 should use 'yeast split ubiquitin assay' rather than Y2H. In the case of the Arp2/3 complex they cited another paper that actually did use Y2H, so it is appropriate to call it Y2H there.

We have now clarified the statements about which exact Y2H assay was used.

As mentioned previously, the citations 7 and 10 do not contain data that supports this statement: "However, standard Y2H and PCA assays are not particularly suitable for the analysis of potentially unstable proteins, as they are barely expressed or rapidly degraded in a cell, resulting in unreliable, mostly false-negative results." Those papers are reviews, and even they do not cite data to support that statement. The authors may make such a statement as a speculation or hypothesis, but if they intend to state it as fact, they must cite primary data to support it.

We have now slightly modified the sentence and removed the references:
"However, standard Y2H and PCA assays may not be well suited for the analysis of potentially unstable proteins. If these proteins are poorly expressed or rapidly degraded in a cell, this will lead to unreliable, false-negative results."

Finally, Supplementary Table 1 in its current form is a little cheesy. First, it introduces a new use for the terms 'direct' and 'indirect', which are different from their definitions throughout the paper.

We have revised the Supplementary Table 1 and no longer use the words "direct" and "indirect" when comparing the readouts of the assays. Instead, we specifically explained the readout.

Second, it combines Y2H and split ubiquitin into the same column and they do not share the same comparative features.

We now use separate columns for the two assays.

Third, in Experimental Setup, is it true that the yeast assays require an incubator and the mammalian cells do not? Or the yeast assay requires a camera and the microscope assays do not? Neither absolutely 'require' a camera as you could just look at the results and record them. These are not noteworthy differences.

We have removed the row with the experimental setup information. We now only highlighted that yeast-based assays require an inexpensive experimental setup.

Fourth, "Possible source for false negative/positive results" appears to allow for speculation, so here is some for ReLo: one or the other protein may locate to the wrong compartment; ectopic overexpression may allow for interaction between proteins that would not normally interact (same for several of the assays listed).

We have now thoroughly revised the Supplementary Table 1. Instead of the categories "advantages/disadvantages" and "possible source of false negative/positive results", we have now included a more general category describing "possible challenges" for each cell-based PPI method.

In the case of the ReLo assay, we have now incorporated some additions. We now highlight that nuclear proteins fused to the PH domain may not completely localize to the plasma membrane, possibly requiring an additional anchoring domain, such as the OST4 miniprotein. We also pointed out (speculated) that interactions between two

proteins that interact through an interface that is embedded in a lipid bilayer cannot be tested.

In response to the reviewer's suggestion, we have clarified for all assays that the possibility of unspecific interactions cannot be ruled out in case of high overexpression.

However, we aim to avoid unnecessary speculation and believe that one suggested source of false positive/negative results in ReLo assays is not justified. The reviewer suggested to add that "one or the other protein may locate to the wrong compartment". While we are a bit unsure about what "wrong" refers to we consider two possibilities:

1. If a protein localizes "wrongly" with respect to its natural localization, it should still form a complex with its coexpressed interaction partner, as long as both interaction partners are present in the cytoplasm and there are no other reasons for a false-negative result, such as steric hindrance due to the fluorescence tag (a reason we have already included in the Supplementary Table 1) or a missing modification. If we think about recombinantly purified proteins, which are also in a "wrong compartment", they form complexes in vitro if they have a certain affinity, and we are therefore unsure about why a "wrong" compartment should prevent an interaction between two proteins.
2. If a protein localizes "wrongly" with respect to the ReLo assay (e.g., not in the cytoplasm), this issue is immediately visible, and the experiment can be adapted, e.g., by deleting strong intrinsic localization signals or testing different anchoring domains. We do not view this as a disadvantage of the ReLo assay per se but rather as a starting point for adapting the assay according to the intrinsic localization of the protein(s) to be tested.

Reviewer #4 (Remarks to the Author):

The authors made a good faith effort to address most of my concerns. I am still not fully convinced it is appropriate for the journal, but the revised version is more scientifically sound.

REVIEWERS' COMMENTS

Reviewer #1 (Remarks to the Author):

The authors tried to do their best in quantifying expression levels by doing western blots with existing primary antibodies. Given the low co-transfection efficiency score, their quantification underlines that their assays rely on expression levels that are far stronger than the endogenous ones. Having a metallothionein promoter for inducible expression would certainly have been more appropriate than the strong constitutive Act5c promoter. Thus, it remains to be known whether this approach could work with normal/endogenous levels of expression.

Reviewer #2 (Remarks to the Author):

The authors are very fortunate to have a devoted professional editor handle this manuscript.

I had three comments in the previous round of review. Two of which dealt with the text/table. The remaining comment suggested that the lack of interaction due to conformational change was not reserved for this ReLo method. The statement seemed to have suggested that the method would allow ones to distinguish conformational changes of proteins to be tested. I understand that reviewer 4 agrees with the authors' claim. If ReLo could reach conclusions that were ambiguous by other PPI assays, such a statement would carry a heavier weight.

The above being said, I did not mean to discourage the journal from publishing this article. By any means, the authors have included robust and solid data in this article aimed at introducing a PPI method.

Reviewer #3 (Remarks to the Author):

Most of the reviewers' concerns have been addressed adequately and this paper now appears ready for publication in Nature Communication.

REVIEWERS' COMMENTS

Reviewer #1 (Remarks to the Author):

The authors tried to do their best in quantifying expression levels by doing western blots with existing primary antibodies. Given the low co-transfection efficiency score, their quantification underlines that their assays rely on expression levels that are far stronger than the endogenous ones. Having a metallothionein promoter for inducible expression would certainly have been more appropriate than the strong constitutive Act5c promoter. Thus, it remains to be known whether this approach could work with normal/endogenous levels of expression.

We intentionally used a strong promoter and believe that the resulting high expression levels are the feature that allows specific detection of direct binary interactions. Due to the excess of ectopically expressed protein, only a small fraction can theoretically be incorporated into endogenous complexes. Our data show that this fraction is insignificant for the assay, as we do indeed detect direct binary interactions. If we were to work at endogenous (i.e. lower) expression levels, the incorporation of ectopically expressed constructs into endogenous complexes (if present) would probably be relatively high and we would probably also detect

interactions mediated by endogenous bridging partners. Therefore, we would not expect a system with endogenous expression levels (or with an inducible promoter) to be more appropriate.

Reviewer #2 (Remarks to the Author):

The authors are very fortunate to have a devoted professional editor handle this manuscript.

I had three comments in the previous round of review. Two of which dealt with the text/table. The remaining comment suggested that the lack of interaction due to conformational change was not reserved for this ReLo method. The statement seemed to have suggested that the method would allow ones to distinguish conformational changes of proteins to be tested. I understand that reviewer 4 agrees with the authors' claim. If ReLo could reach conclusions that were ambiguous by other PPI assays, such a statement would carry a heavier weight.

The above being said, I did not mean to discourage the journal from publishing this article. By any means, the authors have included robust and solid data in this article aimed at introducing a PPI method.

We thank the reviewer for further clarification. We were unsure about the comment regarding the conformations as this comment would actually apply to almost all data that we show. Interaction mapping, mutational analysis, sDMA-dependent interaction testing etc. are all not reserved for ReLo assays. We mostly show validating data to exploit the assay. This is probably why we were confused about the specific statement regarding the conformation testing.

Reviewer #3 (Remarks to the Author):

Most of the reviewers' concerns have been addressed adequately and this paper now appears ready for publication in Nature Communication.